# Sperm membrane proteins DCST1 and DCST2 are required for sperm-egg interaction in mice and fish

Taichi Noda [1,2,3], Andreas Blaha[4,5], Yoshitaka Fujihara [1,6], Krista R. Gert[4,5], Chihiro Emori [1], Victoria E. Deneke[4], Seiya Oura [1,7], Karin Panser[4], Yonggang Lu [1], Sara Berent [4], Mayo Kodani[1,7], Luis Enrique Cabrera-Quio[4,5], Andrea Pauli [4✉] & Masahito Ikawa [1,7,8✉]

The process of sperm-egg fusion is critical for successful fertilization, yet the underlying mechanisms that regulate these steps have remained unclear in vertebrates. Here, we show that both mouse and zebrafish DCST1 and DCST2 are necessary in sperm to fertilize the egg, similar to their orthologs SPE-42 and SPE-49 in *C. elegans* and Sneaky in *D. melanogaster*. Mouse *Dcst1* and *Dcst2* single knockout (KO) sperm are able to undergo the acrosome reaction and show normal relocalization of IZUMO1, an essential factor for sperm-egg fusion, to the equatorial segment. While both single KO sperm can bind to the oolemma, they show the fusion defect, resulting that *Dcst1* KO males become almost sterile and *Dcst2* KO males become sterile. Similar to mice, zebrafish *dcst1* KO males are subfertile and *dcst2* and *dcst1/2* double KO males are sterile. Zebrafish *dcst1/2* KO sperm are motile and can approach the egg, but are defective in binding to the oolemma. Furthermore, we find that DCST1 and DCST2 interact with each other and are interdependent. These data demonstrate that DCST1/2 are essential for male fertility in two vertebrate species, highlighting their crucial role as conserved factors in fertilization.

[1] Research Institute for Microbial Diseases, Osaka University, 3-1 Yamadaoka, Suita, Osaka 565-0871, Japan. [2] Institute of Resource Development and Analysis, Kumamoto University, 2-2-1 Honjo, Chuo-ku, Kumamoto 860-0811, Japan. [3] Priority Organization for Innovation and Excellence, Kumamoto University, 2-39-1 Kurokami, Chuo-ku, Kumamoto 860-8555, Japan. [4] Research Institute of Molecular Pathology (IMP), Vienna BioCenter (VBC), Campus-Vienna-Biocenter 1, 1030 Vienna, Austria. [5] Vienna BioCenter PhD Program, Doctoral School of the University of Vienna and the Medical University of Vienna, 1030 Vienna, Austria. [6] Department of Bioscience and Genetics, National Cerebral and Cardiovascular Center, 6-1 Kishibe-Shimmachi, Suita, Osaka 564-8565, Japan. [7] Graduate School of Pharmaceutical Sciences, Osaka University, 1-6 Yamadaoka, Suita, Osaka 565-0871, Japan. [8] The Institute of Medical Science, The University of Tokyo, 4-6-1 Shirokanedai, Minato-ku, Tokyo 108-8639, Japan. ✉email: andrea.pauli@imp.ac.at; ikawa@biken.osaka-u.ac.jp

Until recently, only a few factors had been shown to be essential for the sperm-egg fusion process: IZUMO1 on the sperm membrane and its receptor (IZUMO1R, also known as JUNO and FOLR4) on the egg membrane (oolemma)[1,2]. Mammalian IZUMO1 and JUNO form a 1:1 complex which is necessary for sperm-egg adhesion prior to fusion[1,3,4]. Furthermore, egg-expressed CD9 is also required for sperm-egg fusion, yet its role appears to be indirect by regulating microvilli formation on the oolemma rather than fusion[5–8]. Recently, we and other research groups have found that four additional sperm factors [fertilization influencing membrane protein (FIMP), sperm-oocyte fusion required 1 (SOF1), transmembrane protein 95 (TMEM95), and sperm acrosome associated 6 (SPACA6)] are also essential for the sperm-egg fusion process and male fertility in mice[9–12]. However, HEK293 cells expressing all of these sperm-expressed, fusion-related factors in addition to IZUMO1 were able to bind but not fuse with zona pellucida (ZP)-free eggs, suggesting that additional fusion-related factors are necessary for the completion of sperm-egg fusion[9].

Dendrocyte expressed seven transmembrane protein (DCSTAMP) and osteoclast stimulatory transmembrane protein (OCSTAMP) represent an interesting group of proteins to study in the context of cell-cell fusion, since they have been shown to play a role in osteoclast and foreign body giant cell (FBGC) fusion[13–15]. They belong to the class of DC-STAMP-like domain-containing proteins and are multi-pass transmembrane proteins with an intracellular C-terminus containing a non-canonical RING finger domain[13,16,17]. DCSTAMP was shown to localize to the plasma membrane and endoplasmic reticulum (ER) membrane in dendritic cells and osteoclasts[16–19]. These cell types in Dcstamp KO mice show no apparent defect in differentiation into the osteoclast lineage and cytoskeletal structure, yet osteoclasts and FBGCs are unable to fuse to form terminally differentiated multinucleated cells[14]. Even though OCSTAMP is widely expressed in mouse tissues[20], the only reported defect in Ocstamp KO mice is the inability to form multinucleated osteoclasts and FBGCs[13,15]. The fusion defect is not due to a change in the expression levels of osteoclast markers, including Dcstamp[13,15]. These results established an essential role for DC-STAMP-like domain-containing proteins in cell–cell fusion.

DC-STAMP-like domain-containing proteins, namely testis-enriched Sneaky, SPE-42, and SPE-49, are necessary for male fertility in Drosophila[21,22] and C. elegans[23–25], respectively. Specifically, sneaky-disrupted fly sperm can enter the egg, but fail to break down the sperm plasma membrane; the male pronucleus thus does not form and embryonic mitotic divisions do not occur[22]. Spe-42 and spe-49 mutant C. elegans sperm can migrate into the spermatheca, the site of fertilization in worms, but these mutants are nearly or completely sterile, respectively, suggesting that SPE-42 and SPE-49 are involved in the ability of sperm to fertilize eggs[23–25]. Sneaky, SPE-42 and SPE-49 have homologs in vertebrates called DC-STAMP domain containing 1 (DCST1) and DCST2, but the roles of these proteins have remained undetermined. Here, we analyzed the physiological function of Dcst1 and Dcst2 and their effect on sperm fertility using genetically modified mice and zebrafish.

## Results

**DCST1 and DCST2 belong to a conserved group of DC-STAMP-like domain-containing proteins**. DC-STAMP-like domain-containing proteins are conserved in metazoa. Phylogenetic analysis revealed a split between the orthologous groups of DCSTAMP and OCSTAMP and of DCST1 and DCST2 (Fig. S1A). Furthermore, DCSTAMP and OCSTAMP as well as DCST1 and DCST2 orthologs form distinct clades, suggesting two gene duplication events at their origin.

Consistent with this, protein sequence identity between the mouse and zebrafish orthologs of DCST1 (39.6% identity) and DCST2 (38.3% identity) is higher than the sequence identity between paralogs (mouse DCST1 and mouse DCST2: 22.5%; zebrafish Dcst1 and zebrafish Dcst2: 21.3%) (Fig. S1B). Based on transmembrane predictions using TMHMM and Phobius[26,27], mouse and zebrafish DCST1 and DCST2 (DCST1/2) have five or six transmembrane helices (Fig. S1C). Their intracellular C-termini contain six invariant cysteines that are thought to form a non-canonical RING finger domain and are required for SPE-42 function in C. elegans[25]. However, the physiological requirements of DCST1/2 in vertebrates have remained unclear.

**DCST1 and DCST2 are required for male fertility in mice**. RT-PCR analysis with multiple mouse tissues showed that Dcst1 and Dcst2 mRNAs are abundantly expressed in mouse testis (Figs. 1A and S2). Using published single-cell RNA-sequencing data[28], we found that Dcst1 and Dcst2 mRNAs peak in mid-round spermatids, indicating that the expression patterns of Dcst1 and Dcst2 are similar to that of other sperm-egg fusion-related genes Izumo1, Fimp, Sof1, and Spaca6 (Fig. 1B).

Using CRISPR/Cas9-mediated mutagenesis, we generated Dcst2 mutant mice lacking 7223 bp (Dcst2$^{del/del}$), which deleted most of the Dcst2 open reading frame (ORF) (Fig. S3A–D). As shown in Fig. S3A, Dcst1 and Dcst2 are adjacent genes in a head-to-head arrangement such that parts of their 5' genomic regions overlap. Because deletion of Dcst2 could in principle affect Dcst1 transcription, we also generated indel mice, Dcst1$^{d1/d1}$ and Dcst2$^{d25/d25}$ (Fig. S4A, B). RNA isolation from mutant testes followed by cDNA sequencing revealed that Dcst1$^{d1/d1}$ has a 1-bp deletion in exon 1, and Dcst2$^{d25/d25}$ has a 25-bp deletion in exon 4 (Figs. S4C, D and S5). Both deletions result in frameshift mutations leading to premature stop codons (Fig. S4E).

Dcst1$^{d1/d1}$, Dcst2$^{d25/d25}$, and Dcst2$^{del/del}$ male mice successfully mated with female mice. However, crosses between Dcst2$^{d25/d25}$ and Dcst2$^{del/del}$ males and wild-type females did not result in any offspring, and crosses with Dcst1$^{d1/d1}$ males were only rarely giving rise to pups {pups/plug: 9.01 ± 2.77 [control (Ctrl), 19 plugs], 0.22 ± 0.19 [Dcst1$^{d1/d1}$, 17 plugs], 0 [Dcst2$^{d25/d25}$, 42 plugs], 0 [Dcst2$^{del/del}$, 24 plugs]}, indicating that Dcst1 mutant males are almost and Dcst2 males are completely sterile (Fig. 1C). Together with our finding that the levels of Dcst1 mRNA were similar between wild-type and the two different Dcst2 mutant testes (Figs. S3D, S4C, and S6), this suggests that (1) the sterility of Dcst2 mutants was caused by the loss of Dcst2 expression and not by a concomitant decrease in Dcst1 expression; and (2) both DCST1 and DCST2 are required for fertilization. Hereafter, we used Dcst1$^{d1/d1}$ and Dcst2$^{d25/d25}$ male mice for all experiments unless otherwise specified.

**Loss of DCST1/2 causes a sperm-egg fusion defect in mice**. The gross morphology of Dcst1$^{d1/d1}$ and Dcst2$^{d25/d25}$ testes was comparable to control heterozygous testes (Fig. S7A). Although the testis weight of Dcst1$^{d1/d1}$ was slightly reduced [testis weight (mg)/body weight (g): 3.13 ± 0.19 (Dcst1$^{d1/wt}$), 2.56 ± 0.27 (Dcst1$^{d1/d1}$), 3.88 ± 0.34 (Dcst2$^{d25/wt}$), 3.60 ± 0.28 (Dcst2$^{d25/d25}$)] (Fig. S7B), PAS-hematoxylin staining revealed no overt defects in spermatogenesis of Dcst1$^{d1/d1}$ and Dcst2$^{d25/d25}$ males (Fig. S7C). The sperm morphology and motility parameters of Dcst1$^{d1/d1}$ and Dcst2$^{d25/d25}$ mice were normal (Fig. S8). However, when mutant sperm were incubated with cumulus-intact wild-type eggs in vitro, they accumulated in the perivitelline space and could not fertilize eggs [96.5 ± 7.1% (Ctrl, 231 eggs), 0% (Dcst1$^{d1/d1}$, 97 eggs), and 0% (Dcst2$^{d25/d25}$, 197 eggs)] (Fig. 1D, E and Movies S1–2). Furthermore, even when these KO sperm were incubated with ZP-free

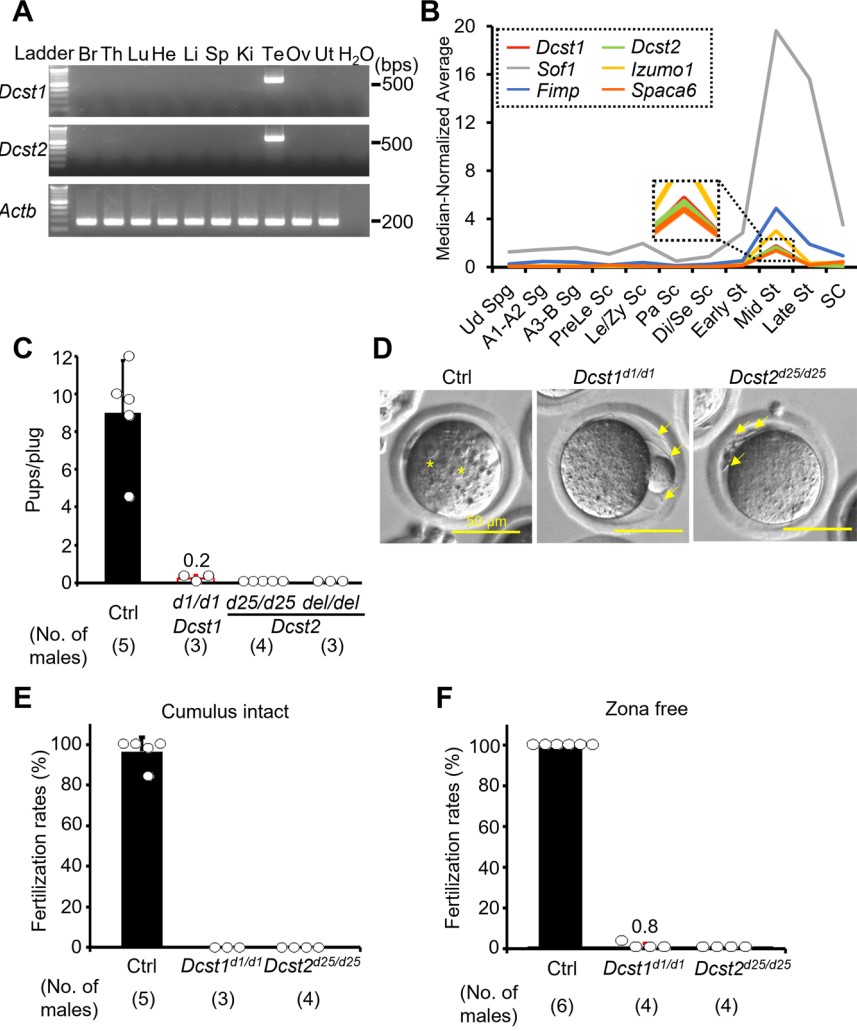

**Fig. 1 Male fertility of *Dcst1* and *Dcst2* mutant mice. A** Multi-tissue gene expression analysis. *Dcst1* and *Dcst2* are abundantly expressed in the mouse testis. Beta actin (*Actb*) was used as the loading control. Br, brain; Th, thymus; Lu, lung; He, heart; Li, liver; Sp, spleen; Ki, kidney; Te, testis; Ov, ovary; Ut, uterus. **B** Median-normalized level of *Dcst1* and *Dcst2* mRNA expression during mouse spermatogenesis. *Dcst1* and *Dcst2* are strongly expressed in mid-round spermatids, corresponding to other fusion-related factors. Ud Sg, undifferentiated spermatogonia; A1-A2 Sg, A1-A2 differentiating spermatogonia; A3-B Sg, A3-A4-In-B differentiating spermatogonia; Prele Sc, preleptotene spermatocytes; Le/Zy Sc, leptotene/zygotene spermatocytes; Pa Sc, pachytene spermatocytes; Di/Se Sc, diplotene/secondary spermatocytes; Early St, early round spermatids; Mid St, mid round spermatids; Late St, late round spermatids; SC, Sertoli cells. **C** Male fecundity. Each male was caged with 2 wild-type females for >1 month. *Dcst2*$^{d25/wt \text{ and } del/wt}$ males were used as the control (Ctrl). *Dcst1*$^{d1/d1}$, *Dcst2*$^{d25/d25}$, and *Dcst2*$^{del/del}$ males succeeded in mating [number of plugs: 19 (Ctrl), 17 (*Dcst1*$^{d1/d1}$), 42 (*Dcst2*$^{d25/d25}$), 24 (*Dcst2*$^{del/del}$)], but the females very rarely delivered pups [pups/plug: 9.0 ± 2.8 (Ctrl), 0.2 ± 0.2 (*Dcst1*$^{d1/d1}$), 0 (*Dcst2*$^{d25/d25}$), 0 (*Dcst2*$^{del/del}$)]. **D** Egg observation after IVF. After 8 h of incubation, pronuclei were observed in the control heterozygous sperm (asterisks). However, *Dcst1* KO and *Dcst2* KO sperm accumulated in the perivitelline space (arrows). The scale bars are 50 μm. **E** Sperm fertilizing ability using cumulus-intact eggs in vitro. *Dcst1* KO and *Dcst2* KO sperm could not fertilize eggs [fertilization rates: 96.5 ± 7.1% (Ctrl, 231 eggs), 0% (*Dcst1*$^{d1/d1}$, 97 eggs), 0% (*Dcst2*$^{d25/d25}$, 197 eggs)]. **F** Sperm fertilizing ability using ZP-free eggs in vitro. *Dcst1* KO and *Dcst2* KO sperm rarely fertilized eggs [fertilization rates: 100% (Ctrl, 142 eggs), 0.8 ± 1.6% (*Dcst1*$^{d1/d1}$, 94 eggs), 0% (*Dcst2*$^{d25/d25}$, 88 eggs)]. All values in this figure are shown as the mean ± SD.

eggs, only one egg and no eggs were fertilized by *Dcst1* KO sperm and *Dcst2* KO sperm, respectively [100% (Ctrl, 142 eggs), 0.8 ± 1.6% (*Dcst1*$^{d1/d1}$, 94 eggs), 0% (*Dcst2*$^{d25/d25}$, 88 eggs)] (Fig. 1F).

To examine the binding and fusion ability of *Dcst1*$^{d1/d1}$ and *Dcst2*$^{d25/d25}$ mutant sperm, we incubated mutant sperm with ZP-free eggs. Both types of mutant sperm could bind to the oolemma [5.72 ± 1.97 (Ctrl, 113 eggs), 7.64 ± 4.68 (*Dcst1*$^{d1/d1}$, 89 eggs), 7.63 ± 3.45 (*Dcst2*$^{d25/d25}$, 89 eggs)] (Fig. 2A). Because binding is not defective in mutant sperm, we confirmed that IZUMO1, a key factor in this process, was expressed normally in testicular germ cells (TGCs) and sperm of *Dcst1*$^{d1/d1}$ and *Dcst2*$^{d25/d25}$ males (Fig. 2B). Indeed, we found that the level of IZUMO1 in mutant sperm was comparable to the control (Figs. 2B and S9). Moreover, the oolemma-bound sperm of *Dcst1*$^{d1/d1}$ and

*Dcst2*$^{d25/d25}$ males underwent the acrosome reaction (AR) normally as determined by live-cell staining with IZUMO1 antibody (Fig. 2C). The number of acrosome-reacted mutant sperm bound to the oolemma was significantly higher than the number of control sperm [3.27 ± 2.31 (Ctrl), 7.34 ± 5.09 (*Dcst1*$^{d1/d1}$), 4.74 ± 2.93 (*Dcst2*$^{d25/d25}$)] (Fig. 2D).

To assess the ability of mutant sperm to fuse with the oolemma, sperm were incubated with Hoechst 33342-preloaded ZP-free eggs. In experiments with control heterozygous sperm, Hoechst 33342 fluorescence signal was translocated into sperm heads (Fig. 2E, arrow), indicating that these sperm fused with the egg membrane. However, Hoechst 33342 signal was rarely detected in *Dcst1* KO and not detected in *Dcst2* KO sperm bound to the oolemma [fused sperm/egg: 1.52 ± 0.35 (Ctrl, 113 eggs), 0.04 ± 0.05 (*Dcst1*$^{d1/d1}$, 73

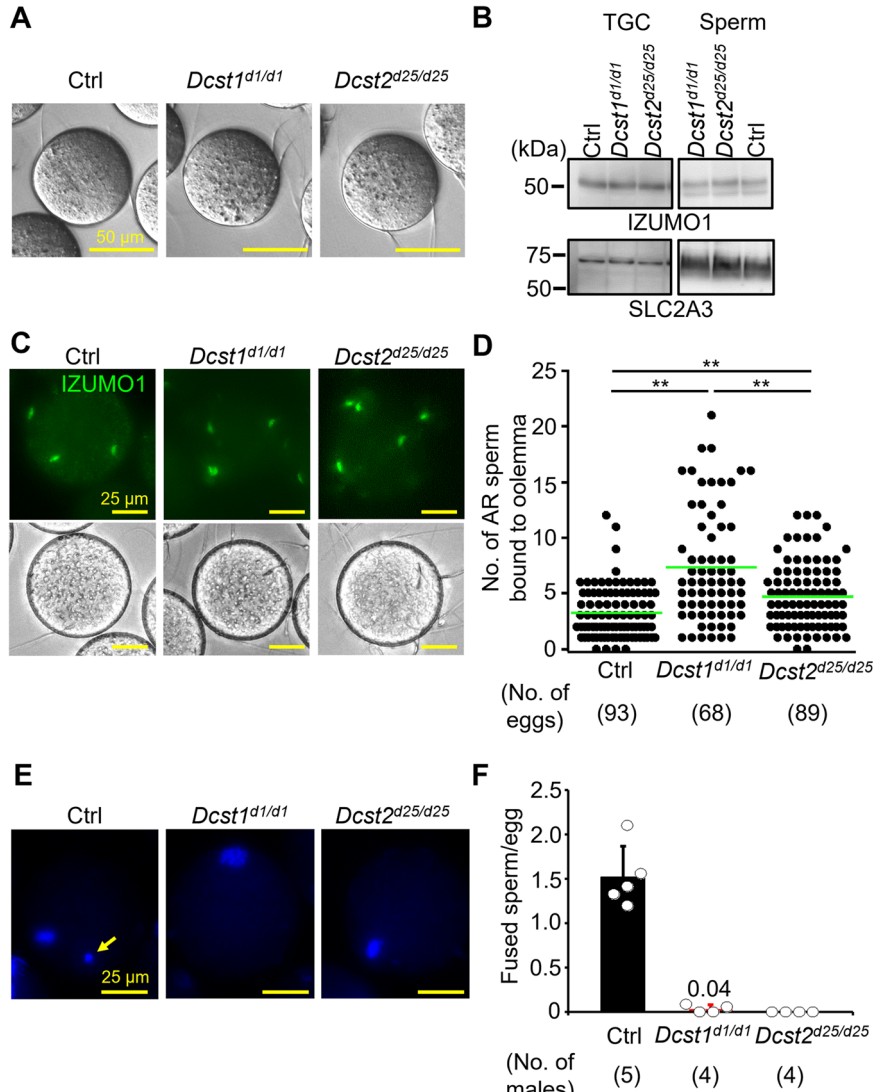

**Fig. 2 Adhesion and fusion ability of *Dcst1* and *Dcst2* mutant sperm to oocyte plasma membrane. A** Binding ability. *Dcst1* KO and *Dcst2* KO sperm could bind to the oolemma after 30 min of incubation. The scale bars are 50 µm. **B** Detection of IZUMO1. The band signals of IZUMO1 in TGC and sperm of *Dcst1^{d1/d1}* and *Dcst2^{d25/d25}* male mice were comparable to the control wild-type sperm. SLC2A3, one of proteins in sperm tail, was used as the loading control. **C**, **D** Acrosome status of binding sperm. Live sperm bound to the oolemma were stained with the IZUMO1 antibody, and IZUMO1 only in the acrosome reacted (AR) sperm was detected. The AR-sperm of both *Dcst* mutants could bind to the oolemma, and their numbers were higher than the control heterozygous sperm [3.27 ± 2.31 (Ctrl, 93 eggs), 7.34 ± 5.09 (*Dcst1^{d1/d1}*, 68 eggs), 4.74 ± 2.93 (*Dcst2^{d25/d25}*, 89 eggs)]. **p < 0.01. The scale bars are 25 µm. **E**, **F** Fusion ability. The ZP-free eggs pre-stained Hoechst 33342 were used for sperm-egg fusion assay. Hoechst 33342 signal transferred to control sperm heads, indicating that sperm fused with eggs (panel **E**, arrow). However, *Dcst1* KO and *Dcst2* KO sperm rarely fused with eggs [fused sperm/egg: 1.52 ± 0.35 (Ctrl, 113 eggs), 0.04 ± 0.05 (*Dcst1^{d1/d1}*, 73 eggs), 0 (*Dcst2^{d25/d25}*, 73 eggs)]. The scale bars are 25 µm. All values are shown as the mean ± SD.

eggs), 0 (*Dcst2^{d25/d25}*, 73 eggs)] (Fig. 2E, F). These results indicate that control heterozygous sperm can fuse with eggs but *Dcst2* KO and *Dcst1* KO sperm are impaired at the step of fusion (Fig. 2E). The fusion defect is in agreement with the increased number of sperm bound to the oolemma due to the absence of the membrane block of polyspermy that is normally triggered by fertilization[1]. Thus, while acrosome-reacted sperm of *Dcst1* and *Dcst2* KO mice can bind to eggs, they are defective at fusing with the oolemma: KO males of *Dcst1* or *Dcst2* causes a strong impairment or complete loss of sperm-egg fusion, respectively.

**Fecundity of *Dcst1^{d1/d1}* and *Dcst2^{d25/d25}* males is rescued by *Dcst1*-3xHA and *Dcst2*-3xHA transgenes.** To confirm that the

*Dcst1* and *Dcst2* disruptions are responsible for male sterility, we generated transgenic mice in which a testis-specific Calmegin (*Clgn*) promoter expresses mouse DCST1 and DCST2 with an HA tag at the C-terminus (Fig. S10A, B). When *Dcst1^{d1/d1}* males with the *Dcst1*-3xHA transgene and *Dcst2^{d25/d25}* males with the *Dcst2*-3xHA transgene were mated with wild-type females, the females delivered normal numbers of offspring [pups/plug: 5.7 ± 0.5 (*Dcst1^{d1/d1}*; Tg, 25 plugs), 7.6 ± 2.7 (*Dcst2^{d25/d25}*; Tg, 15 plugs)] (Fig. 3A). Although both transgenes code for a C-terminal HA-tag and could rescue *Dcst1/2* KO, only very low levels of HA-tagged DCST1 could be detected in sperm samples (Fig. S10C), suggesting that only low levels of DCST1 are required for fertilization. HA-tagged DCST2 was detected in TGCs and sperm at the expected size for the full-length protein (Figs. 3B and S10C,

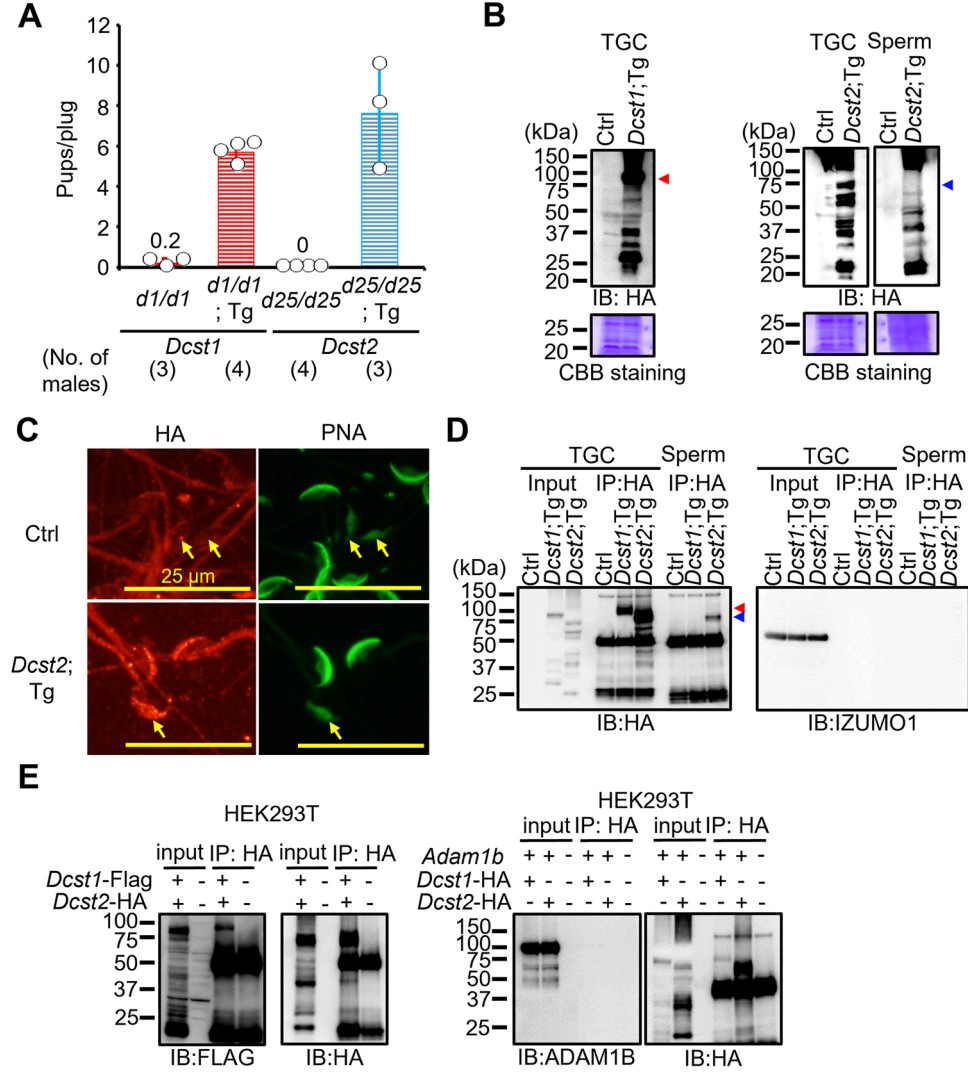

**Fig. 3 Detection of DCST1/2 in TGC and sperm and interaction of DCST1/2. A** Rescue of male fertility. *Dcst1*$^{d1/d1}$ males with *Dcst1*-3xHA Tg insertion and *Dcst2*$^{d25/d25}$ males with *Dcst2*-3xHA Tg insertion were generated (Fig. S10), and their fertility was rescued [number of plugs: 17 (*Dcst1*$^{d1/d1}$), 25 (*Dcst1*$^{d1/d1}$;Tg), 42 (*Dcst2*$^{d25/d25}$), 15 (*Dcst2*$^{d25/d25}$;Tg)]. The fecundity data in *Dcst1*$^{d1/d1}$ and *Dcst2*$^{d25/d25}$ males is replicated from Fig. 1C. All values are shown as the mean ± SD. **B** Detection of DCST1 and DCST2 in TGC and sperm. The protein extract of TGC (100 µg) and sperm (6.6 × 10^6 sperm) was used for SDS-PAGE. The HA-tagged DCST1 and HA-tagged DCST2 were detected in TGC and sperm. Total proteins in the membrane were visualized by CBB staining. Triangle marks show the expected molecular size of DCST1 (about 80 kDa) and DCST2 (about 77 kDa). **C** Localization of DCST2 in sperm. The HA-tagged DCST2 was localized in the anterior acrosome before the acrosome reaction, and then translocated to the equatorial segment in acrosome-reacted sperm (arrows). Wild-type sperm were used as the negative control. PNA was used as a marker for the acrosome reaction. The fluorescence in the sperm tail was non-specific. The scale bars are 25 µm. **D** Co-IP and western blotting of the interaction between IZUMO1 and DCST1/2. The TGC and sperm lysates from Ctrl (*Dcst2*$^{wt/wt}$ and $^{d25/wt}$ mice), *Dcst1*;Tg, and *Dcst2*;Tg males were incubated with anti-HA tag antibody-conjugated magnetic beads, and then the eluted protein complex was subjected to western blotting. The HA-tagged DCST1 was detected only in the IP product from TGC, and the HA-tagged DCST2 was detected in the IP-product from TGC and sperm. IZUMO1 was not detected in the co-IP products. Red and blue triangle marks show the expected molecular size of DCST1 (about 80 kDa) and DCST2 (about 77 kDa), respectively. **E** Interaction between DCST1 and DCST2 in HEK293T cells. The protein lysate collected from HEK cells overexpressing *Dcst1*-3xFLAG and *Dcst2*-3xHA was incubated with anti-HA tag antibody-conjugated magnetic beads. The FLAG-tagged DCST1 was detected in the eluted protein complex. ADAM1B, a sperm protein that localizes to the sperm surface and is not involved in sperm-egg fusion, was used for negative control.

arrowheads), though both DCST1 and DCST2 proteins appear to be subject to post-translational processing or protein degradation.

To reveal the localization of DCST2 in sperm, we performed immunocytochemistry with an antibody detecting the HA epitope and peanut agglutinin (PNA) as a marker for the sperm acrosome reaction. As shown in Fig. 3C, PNA in the anterior acrosome was translocated to the equatorial segment after the acrosome reaction as shown previously[29]. HA-tagged DCST2 was detected within the anterior acrosome of acrosome-intact sperm, and then translocated to the equatorial segment after the

acrosome reaction (Fig. 3C), mirroring the relocalization of IZUMO1 upon the acrosome reaction[30]. Fluorescence in the sperm tail was observed in both control wild-type and *Dcst2-HA* Tg sperm, indicating that this signal in the tail was non-specific.

Taking advantage of the HA tag, we performed co-immunoprecipitation (co-IP). While HA-tagged DCST1 was detected only in TGCs, HA-tagged DCST2 was detected in both TGCs and sperm (Fig. 3D). We could not detect IZUMO1 in these IP samples (Fig. 3D), suggesting that DCST1 and DCST2 do not form a complex with IZUMO1. However, co-expression of

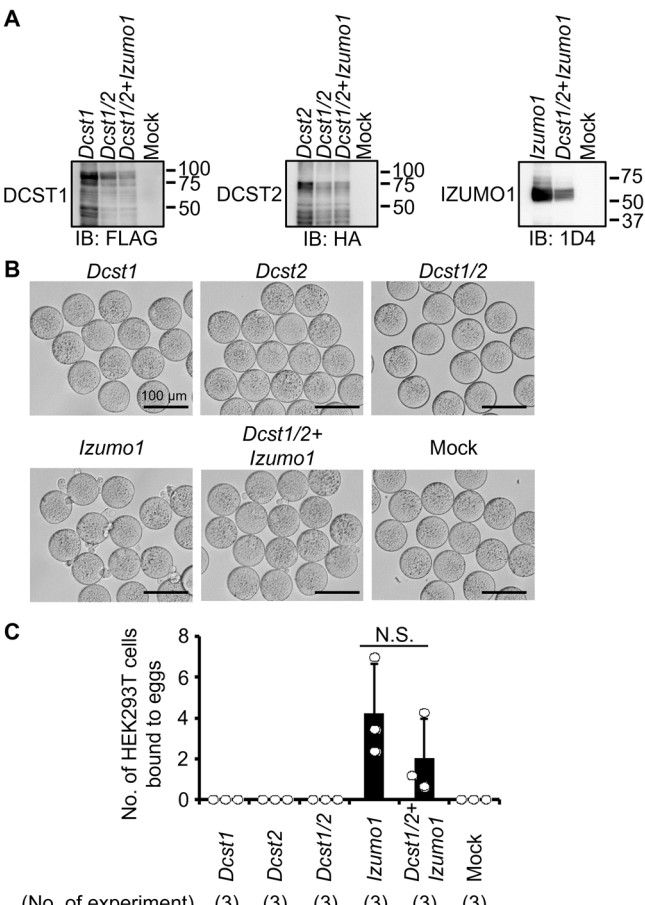

**Fig. 4 Binding assay between ZP-free eggs and HEK293T cells overexpressing Dcst1/2. A** Detection of DCST1/2 and IZUMO1. FLAG-tagged DCST1, HA-tagged DCST2, and 1D4-tagged IZUMO1 were detected in HEK293T cells overexpressing *Dcst1*-3xFLAG, *Dcst2*-3xHA, and *Izumo1*-1D4. **B, C** Observation of ZP-free eggs incubated with HEK293T cells overexpressing *Dcst1/2* and *Izumo1*. The HEK293T cells overexpressing *Dcst1* or *Dcst2* did not attach to the oocyte membrane. Even when the HEK cells overexpressing *Dcst1/2* were used for the assay, these cells failed to bind to ZP-free eggs. The HEK293T cells overexpressing *Dcst1/2* and *Izumo1* could bind to the oocyte membrane but could not fuse with an egg. n.s.: not significant ($p = 0.28$). The scale bars are 100 μm. All values are shown as the mean ± SD.

*Dcst1*-3xFLAG and *Dcst2*-3xHA in HEK293T cells revealed the presence of a DCST1/DCST2 complexes (Fig. 3E), which is in line with proposed complex formation between OCSTAMP and DCSTAMP during osteoclast fusion[31].

**HEK293T cells expressing DCST1/2 and IZUMO1 bind to, but do not fuse with, ZP-free eggs.** To assess whether DCST1 and DCST2 are sufficient for inducing sperm-egg fusion, we overexpressed *Dcst1*-3xFLAG, *Dcst2*-3xHA, and *Izumo1*-1D4 in HEK293T cells (Figs. 4A and S11). HEK293T cells overexpressing IZUMO1 could bind to, but not fuse with, ZP-free eggs (Fig. 4B), which was consistent with previous reports[4,9]. In contrast, HEK293T cells overexpressing only DCST1 and DCST2 failed to bind to ZP-free eggs (Fig. 4B, C). Co-expression of IZUMO1 and DCST1/2 allowed the cells to bind to ZP-free eggs [4.24 ± 2.41 cells/eggs (IZUMO1), 2.01 ± 1.93 cells/eggs (DCST1/2 and IZUMO1)], but did not facilitate fusion with the oolemma (Fig. 4B, C). Thus, though DCST1 and DCST2 appear to have a role in the sperm-egg

fusion process, they are not sufficient to induce fusion, even in conjunction with IZUMO1.

**Sperm-expressed Dcst1/2 are also required for fertilization in zebrafish.** To assess to what extent our findings in mice could be expanded among vertebrate species, we asked what the roles of DCST1/2 are in an evolutionarily distant vertebrate species, the zebrafish. The orthologous zebrafish genes *dcst1* and *dcst2* are expressed specifically in testis and arranged similarly to mouse *Dcst1/2* (Fig. S12A, B). We therefore generated three independent KO fish lines, *dcst1*$^{-/-}$, *dcst2*$^{-/-}$, and *dcst1/2*$^{-/-}$, by CRISPR/Cas9-mediated mutagenesis (Fig. S12B, C). Lack of zebrafish Dcst2 alone or in combination with Dcst1 caused complete sterility in males, whereas lack of Dcst1 alone led to severe sub-fertility [5.5 ± 3.6% fertilization rate (*dcst1*$^{-/-}$, 16 clutches)] (Fig. 5A). The fertility of heterozygous males and KO females, however, was comparable to wild-type control males. (Fig. 5A). Thus, similar to mice, Dcst1/2 are essential for male fertility in zebrafish.

To understand what causes the fertility defect, we first determined whether sperm were produced in mutant males. *Dcst1*$^{-/-}$, *dcst2*$^{-/-}$, and *dcst1/2*$^{-/-}$ males showed normal mating behavior and produced morphologically normal sperm (DIC images in Fig. 5B), indicating that zebrafish Dcst1/2 are not crucial for spermatogenesis. To detect Dcst1 and Dcst2 proteins, we produced antibodies against the C-terminal RING finger domains of zebrafish Dcst1 and Dcst2. Each antibody was specific against its cognate target antigen as determined by western blotting of wild-type and KO sperm lysates (Fig. 5C), and Dcst2 was detected by immunofluorescence staining of zebrafish embryos overexpressing *dcst2(RING)-superfolder GFP (sfGFP)* mRNA (Fig. S12D). Interestingly, *dcst1*$^{-/-}$ and *dcst2*$^{-/-}$ sperm were not only lacking Dcst1 or Dcst2 protein, respectively, but were lacking both Dcst proteins (Fig. 5C). This suggests that Dcst1/2 protein stability requires the presence of both proteins, which is consistent with mouse DCST1/2 forming a protein complex (Fig. 3E). To examine where Dcst2 is localized in zebrafish sperm, which lacks an acrosome, we performed immunofluorescence against Dcst2. Wild-type sperm was strongly stained at the periphery of the head in punctae and occasionally the mid-piece (Fig. 5B). Staining of the mid-piece and occasionally the tail region was also detected in *dcst2* KO sperm, suggesting that this signal was unrelated to Dcst2. Dcst2 foci are markedly reduced in *dcst1*$^{-/-}$ sperm, which is consistent with the observed interdependence of Dcst1/2 in sperm lysates.

When added to wild-type eggs, *dcst2* KO sperm were able to reach and enter the micropyle, the funnel-shaped site of sperm entry, similar to wild-type sperm (Fig. 5D; Movies S3 and S4). We therefore conclude that Dcst2 is neither required for overall sperm motility nor for sperm to approach and enter the micropyle. However, in contrast to wild-type sperm which remained attached to the egg (Movie S3), most of the entering mutant sperm subsequently detached and drifted away from the micropyle (Movie S4), suggesting that sperm lacking Dcst2 are defective in stable binding to the oolemma. We previously established an assay to assess sperm-egg binding during zebrafish fertilization[32]. Building on this assay, we used live imaging of sperm and eggs to quantify the number of wild-type sperm adhered to the oolemma within a physiologically relevant time frame [1.97 ± 0.97 sperm/100 μm (12 eggs)] (Fig. 5E, F and Movie S5). Performing this assay with *dcst2* KO sperm revealed that these sperm were unable to adhere stably to wild-type eggs [0.05 ± 0.1 sperm/100 μm (9 eggs)] (Fig. 5E, F and Movie S5). We therefore conclude that zebrafish Dcst2 is required for stable binding of sperm to the oolemma.

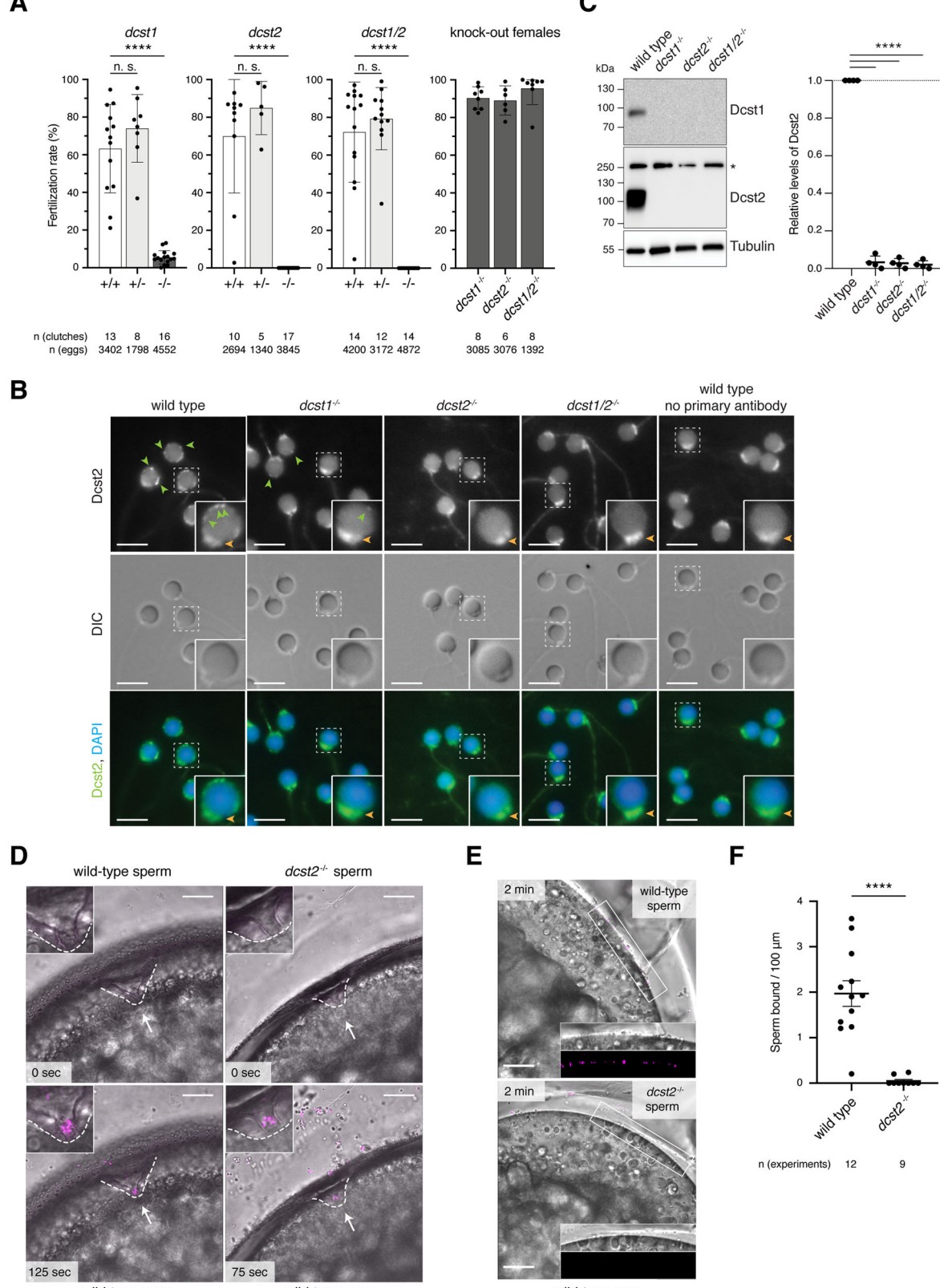

## Discussion

Here, we demonstrate that the testis-enriched proteins DCST1/2 are necessary for male fertility in mice and fish. These results agree with the recently reported independent findings in mice[33] and the known requirements of *Drosophila* Sneaky and *C. elegans* SPE-42/49 in sperm for successful reproduction[21–25], indicating that the physiological function of DCST1/2 is widely conserved among bilaterians. The conservation of DCST1/2 outside of vertebrates is exceptional not only because fertilization-relevant genes are rapidly evolving[34,35] but also because other known vertebrate proteins involved in sperm-egg interaction have orthologs only in select vertebrate genera[36–39]. Contrary to vertebrates, which have

**Fig. 5 Dcst1 and dcst2 are essential for male fertility in zebrafish. A** Dcst1 and dcst2 mutant zebrafish are male sterile. Quantification of fertilization rates as assessed by the number of embryos that progress beyond the one-cell stage. Left three panels: Males of different genotypes [wild-type sibling (+/+; white); heterozygote sibling (+/−; light gray); homozygote sibling (−/−; dark gray)] were crossed to wild-type females; right panel: homozygous mutant females (−/−; dark gray) of the indicated genotypes were crossed to wild-type males. The number of individual clutches and the total number of eggs per genotype are indicated. Data are means ± SD; adj. ****$p < 0.0001$ (Kruskal–Wallis test with Dunn's multiple-comparisons test); n.s., not significant. **B** Dcst2 localizes to the periphery of the sperm head. Immunofluorescent and differential interference contrast (DIC) images of wild-type and mutant zebrafish sperm. Sperm were stained with DAPI (blue) to visualize nuclei and an antibody against the RING-finger domain of zebrafish Dcst2. Dcst2 localizes to distinct foci around the sperm head (green arrowheads). Dcst2 foci are reduced ($dcst1^{−/−}$) or not detectable ($dcst2^{−/−}$; $dcst1/2^{−/−}$) in mutant sperm, while overall sperm morphology in DIC images appears normal for all genotypes. Autofluorescence of the sperm mid-piece appears as a uniform signal in the Dcst2 channel (orange arrowhead) in all genotypes. Scale bar: 5 μm. Boxed inset shows an individual enlarged sperm head for each genotype. **C** Dcst1 and Dcst2 are absent in mutant sperm. Exemplary immunoblot of sperm samples of wild-type and mutant genotypes probed with antibodies against zebrafish Dcst1 and Dcst2. Tubulin protein levels of the same blot are shown as loading control. For Dcst2, an unspecific band (asterisk) is detected in all genotypes at ~250 kDa. Quantification of Dcst2 levels of wild-type and mutant sperm based on $n = 4$ biologically independent immunoblots. Values were normalized to tubulin and then to wild-type levels. Data are means ± SD; ****$p < 0.0001$ (one-way ANOVA and multiple comparisons analysis). **D** Dcst2 mutant sperm are motile and reach the micropyle. Images from time-lapse movies of wild-type (left) or $dcst2^{−/−}$ (right) sperm added to wild-type eggs. Sperm (magenta) were labeled with MitoTracker and added to inactivated eggs. Sperm and eggs were activated by addition of water just before the start of the movie. The micropyle (white arrow), a preformed funnel in the egg coat through which the sperm reach the oolemma, is outlined with a dashed white line. Top images depict the first acquired image following sperm addition: no sperm has entered the micropylar area in either wild-type or mutant samples. Bottom images (125 and 75 s after sperm addition in wild-type and $dcst2^{−/−}$ samples, respectively): sperm can readily be detected within the micropylar area (inset). Scale bar: 75 μm. **E, F** Dcst2 mutant sperm are defective in stable binding to wild-type eggs. **E** Images from a time-lapse movie of wild-type (top) or $dcst2^{−/−}$ (bottom) sperm added to activated and dechorionated wild-type eggs. Sperm (magenta) were labeled with MitoTracker and activated at the time of addition to the eggs. Wild-type sperm show clear binding to the surface of the egg (inset), while $dcst2^{−/−}$ sperm are unable to stably bind to the oolemma. Scale bar: 50 μm. **F** Binding of sperm was assessed by quantifying the number of stably bound sperm in a 1-min time window. The number of independent experiments is indicated. ****$p < 0.0001$ (Mann–Whitney test).

both DCST1/2 and the related DCSTAMP/OCSTAMP proteins (Fig. S1A), invertebrates have solely DCST1/2.

Having established the essential role of DCST1/2 for male fertility, questions arise concerning the molecular processes in which they are involved and how they contribute to fertilization. We detected mouse DCST2 at the equatorial segment of acrosome-reacted sperm and zebrafish Dcst2 at the periphery of the sperm head. Since DCST1/2 are TM proteins, these localization patterns suggest that part of the proteins are exposed on the sperm surface. Given our result that DCST1/2 form a complex (Fig. 3E) and are interdependent of each other (Fig. 5C), we speculate that a DCST1/2 complex in the sperm membrane helps organize the presentation of other fusion-related sperm proteins. The recently reported absence of SPACA6 in Izumo1 or Dcst1/2 KO sperm supports this idea[33]. Another possibility is that DCST1/2 directly interact with other binding- and/or fusion-relevant molecules on the oolemma. To test this hypothesis, we overexpressed DCST1 and DCST2 in IZUMO1-expressing HEK293T cells. These cells could bind to, but not fuse with, ZP-free eggs (Fig. 4B, C), suggesting that DCST1/2 are not sufficient to induce fusion in a heterologous system. This observed lack of fusion could be due to the absence of other sperm-oocyte fusion-related factors (FIMP, SOF1, TMEM95, and SPACA6). Future investigation is needed to uncover the mechanism by which DCST1/2 act during fertilization.

The role of DCST1/2 during sperm-egg interaction differs between mice and fish. Specifically, loss of mouse DCST1/2 led to a defect in sperm-egg fusion (Fig. 2), suggesting that DCST1/2 may directly or indirectly regulate membrane fusion via an IZUMO1-independent pathway or act as fusion mediators downstream of the interaction between IZUMO1 and JUNO. Interestingly, zebrafish Dcst1/2 are required for sperm-egg binding (Fig. 5). Given the diversity of the fertilization process across the animal kingdom, it may be that while DCST proteins are widely conserved[36,37], they may have evolved different roles to fit into the specific context of fertilization for a given species or genus. Application of CRISPR/Cas9-mediated KO technology on a larger set of animals with divergent modes of fertilization will shed light on both the conservation and possible species-specific diversification in DCST1/2's function in fertilization.

## Materials and methods

**Animals**. B6D2F1, C57BL/6J, and ICR mice were purchased from Japan SLC and CLEA Japan. Mice were acclimated to 12-h light/12-h dark cycle. All animal experiments were approved by the Animal Care and Use Committee of the Research Institute for Microbial Diseases, Osaka University, Japan (#Biken-AP-H30-01) and the Animal Care and Use Committee of Kumamoto University (ID: A2021-035). Experiments were performed with mice between 3 weeks and 10 months of age.

Zebrafish (Danio rerio) were raised according to standard protocols (28 °C water temperature; 14/10-h light/dark cycle). TLAB zebrafish served as wild-type zebrafish for all experiments and were generated by crossing zebrafish AB stocks with natural variant TL (Tüpfel longfin) stocks. $Dcst1^{−/−}$, $dcst2^{−/−}$, and $dcst1/2^{−/−}$ mutant zebrafish were generated as part of this study as described in detail below. All experiments were conducted with 3-month- to 1.5-year-old fish according to Austrian and European guidelines for animal research and approved by the local Austrian authorities (animal protocol GZ: 342445/2016/12).

**Mouse sample collection**. The brain, thymus, lung, heart, liver, spleen, kidney, testis, ovary, and uterus were collected from C57BL/6J mice. For western blotting, TGC proteins were extracted with Pierce IP lysis buffer (Thermo Fisher Scientific) (Fig. 3B, D) or RIPA buffer [50 mM Tris HCl (pH 7.5), 0.15 M NaCl, 1% Sodium deoxycholate, 0.1% SDS, 1% (vol/vol) TritonX-100] containing a 1% (vol/vol) protease inhibitor mixture (Nacalai Tesque) (Fig. 2B). Proteins of cauda epididymal sperm were extracted with Pierce IP lysis buffer containing a 1% (vol/vol) protease inhibitor mixture (Fig. 3D) or SDS sample buffer containing β-mercaptoethanol (Nacalai Tesque) (Figs. 2B, 3B) as described previously[40].

**Reverse transcription polymerase chain reaction (RT-PCR) for mouse multi-tissue expression analyses**. Mouse tissues were homogenized in TRIzol reagent (Thermo Fisher Scientific), and total RNA was extracted as described in the instruction manual. Total RNA was reverse-transcribed into cDNA using a SuperScript IV First-Strand Synthesis System (Invitrogen) and Deoxyribonuclease (RT Grade) (Nippon gene). PCR was conducted with primer sets (Table S1) and KOD-Fx neo (TOYOBO). The PCR conditions were initial denaturation at 94 °C for 3 min, denaturing at 94 °C for 30 s, annealing at 65 °C for 30 s, and elongation at 72 °C for 30 s for 35 cycles in total, followed by 72 °C for 2 min.

**Single cell RNA-seq (scRNAseq) analysis**. The Median-Normalized average of Dcst1, Dcst2 and fusion-related genes (Fimp, Izumo1, Sof1, and Spaca6) in spermatogenesis was examined in the published scRNAseq database[28].

**Mouse mating test**. KO male mice were caged with two B6D2F1 females for more than 1 month. After the mating period, male mice were removed from the cages, and the females were kept for another 20 days to allow them to deliver offspring. Frozen sperm from $Dcst1^{d1/wt}$ males [B6D2-Dcst1 < em2Osb > , RBRC#10332, CARD#2702], $Dcst2^{d25/wt}$ males [B6D2-Dcst2 < em2Osb > Tg(CAG/Su9-DsRed2,Acr3-

EGFP)RBGS002Osb, RBRC#11243, CARD#3047], Dcst2$^{del/wt}$ males [B6D2-Dcst2 < em3Osb > , RBRC#11489, CARD#3133], Dcst1$^{d1/d1}$; Tg males [B6D2-Dcst1 < em2Osb > Tg(Clgn-Dcst1/3xHA)1Osb, RBRC#11491, CARD#3135], and Dcst2$^{d25/d25}$; Tg males [B6D2-Dcst2 < em2Osb > Tg(Clgn-Dcst2/3xHA)1Osb, RBRC#11488, CARD#3132] will be available through RIKEN BRC (http://en.brc.riken.jp/index.shtml) and CARD R-BASE (http://cardb.cc.kumamoto-u.ac.jp/transgenic/).

**Mouse sperm motility and in vitro fertilization.** Cauda epididymal sperm were squeezed out and dispersed in PBS (for sperm morphology) and TYH (for sperm motility and IVF)[41]. After incubation of 10 and 120 min in TYH, sperm motility patterns were examined using the CEROS II sperm analysis system[42–44]. IVF was conducted as described previously[45]. Sperm of Dcst1$^{d1/wt}$ and Dcst2$^{d25/wt}$ mice were used as the control for sperm motility and IVF. Protein extracts from the remaining sperm suspension in PBS and TYH drops were used for co-IP experiments.

**Antibodies.** Rat monoclonal antibodies against mouse IZUMO1 (KS64-125) and mouse SLC2A3 (KS64-10) were generated by our laboratory as described previously[46,47]. The mouse monoclonal antibody against 1D4-tag was generated using a hybridoma cell line as a gift from Robert Molday, Ophthalmology and Visual Sciences, Centre for Macular Research, University of British Columbia, Vancouver, British Columbia, Canada[48]. Mouse monoclonal antibodies against the HA and FLAG tags were purchased from MBL (M180-3) and Sigma (F3165). The Alexa Fluor 488-conjugated Lectin PNA from Arachis hypogaea (peanut) was purchased from Thermo Fisher Scientific (L21409).

To generate mouse monoclonal antibodies against zebrafish Dcst1 and Dcst2, recombinant zebrafish Dcst1 (amino acids 590–675) and Dcst2 (amino acids 574–709) proteins were expressed in E. coli BL21(DE3) and purified by the VBCF Protein Technologies Facility. Each recombinant protein was injected into 3 mice, and monoclonal antibodies were generated by the Max Perutz Labs Monoclonal Antibody Facility according to standard procedures.

Horseradish peroxidase (HRP)-conjugated goat anti-mouse immunoglobulins (IgGs) (115-036-062) and HRP-conjugated goat anti-rat IgGs (112-035-167) were purchased from Jackson ImmunoResearch Laboratories. Fluorophore-conjugated secondary antibodies, goat anti-mouse IgG Alexa Fluor 488 (A11001), goat anti-mouse IgG Alexa Fluor 546 (A11018), goat anti-mouse IgG Alexa Fluor 594 (A11005), and goat anti-rat IgG Alexa Fluor 488 (A11006) were purchased from Thermo Fisher Scientific.

**Mouse sperm-egg binding and fusion assay.** The sperm-egg binding and fusion assay was performed as described previously[9]. To visualize IZUMO1 distribution in sperm, sperm after incubation of 2.5 h in TYH drops were then incubated with the IZUMO1 monoclonal antibody (KS64-125, 1:100) for 30 min. Then, sperm were incubated with ZP-free eggs in TYH drops with the mixture of IZUMO1 monoclonal antibody (KS64-125, 1:100) and goat anti-rat IgG Alexa Fluor 488 (1:200) for 30 min. Then, the eggs were gently washed with a 1:1 mixture of TYH and FHM medium three times, and then fixed with 0.2% PFA. After washing again, IZUMO1 localization was observed under a fluorescence microscope (BZ-X700, Keyence). Sperm of Dcst1$^{d1/wt}$ and Dcst2$^{d25/wt}$ mice were used as the control.

**Western blotting.** Mouse samples were mixed with sample buffer containing β-mercaptoethanol[40], and boiled at 98 °C for 5 min prior to SDS PAGE. After protein transfer, the polyvinylidene difluoride (PVDF) membrane was treated with Tris-buffered saline (TBS)-0.1% Tween20 (Nacalai Tesque) containing 10% skim milk (Becton Dickinson and Company) for 1 h, followed by the primary antibody [IZUMO1, SLC2A3, HA, and FLAG (1:1,000), 1D4 (1:5,000)] for 3 h or overnight. After washing with TBST, the membrane was treated with secondary antibodies (1:1,000). The HRP activity was visualized with ECL prime (BioRad) and Chemi-Lumi One Ultra (Nacalai Tesque). Then, the total amount of proteins on the membrane was visualized with Coomassie Brilliant Blue (CBB) (Nacalai Tesque).

Zebrafish sperm from 3 to 6 males was sedimented at 3000 rpm for 3.5 min. The supernatant was replaced with modified RIPA buffer [50 mM Tris-HCl (pH 7.5), 150 mM NaCl, 1 mM MgCl₂, 1% NP-40, 0.5% sodium deoxycholate, 1% SDS, 1× complete protease inhibitor (Roche)]. After preparation of all samples, 1 U/μl benzonase (Merck) was added, and samples were incubated for 30 min at room temperature. Samples were then mixed with semi-native SDS-PAGE sample buffer [100 mM Tris-HCl (pH 6.8), 8% glycerol, 0.1 mg/ml bromophenol blue, 2% SDS, 10 mM DTT][49] and heated to 42 °C for 5 min, or alternatively, mixed with 4× Laemmli containing β-mercaptoethanol and heated at 95 °C for 5 min. After SDS-PAGE, samples were wet-transferred onto a nitrocellulose membrane. Total protein was visualized by Ponceau staining before blocking with 5% milk powder in 0.1% Tween in 1x TBS (TBST). Membranes were incubated in primary antibody overnight at 4 °C, then washed with TBST before HRP-conjugated secondary antibody [1:10.000 (115-036-062, Dianova)] incubation for 1 h. Membranes were washed several times in TBST before HRP activity was visualized using Clarity Western ECL Substrate (BioRad) on a ChemiDoc (BioRad). After detection, membranes were stripped using Restore Western Blot Stripping Buffer (Thermo Fisher Scientific) before re-blocking and proceeding with another primary antibody incubation. Primary antibodies: mouse anti-zebrafish-Dcst1 (1:25); mouse anti-zebrafish-Dcst2 (1:500); mouse anti-alpha-Tubulin [1:20.000 (T6074, Merck)]. To

assess relative Dcst2 protein amounts, average intensities of Dcst2-specific bands were quantified in Fiji on 4 independent immunoblots for each genotype relative to Tubulin levels, which was used as loading control. Values were then normalized to the levels of wild-type sperm.

**Immunocytochemistry.** After 3-h incubation of mouse sperm in TYH drops, sperm were washed with PBS. Sperm suspended with PBS were smeared on a slide glass, and then dried on a hotplate. The samples were fixed with 1% PFA, followed by permeabilization with Triton-X 100. Sperm were blocked with 10% goat serum (Gibco) for 1 h, and then incubated with a mouse monoclonal antibody against HA tag (1:100) for 3 h or overnight. After washing with PBS containing 0.05% (vol/vol) Tween 20, the samples were subjected to the mixture of a goat anti-mouse IgG Alexa Fluor 546 (1:300) and Alexa Fluor 488-conjugated Lectin PNA (1:2000) for 1 h. After washing again, the samples were sealed with Immu-Mount (Thermo Fisher Scientific) and then observed under a phase contrast microscope (BX-50, Olympus) with fluorescence equipment.

Zebrafish sperm was fixed with 3.7% formaldehyde diluted in Hank's saline immediately after collection and stored on ice for 20 min to 1 h. Sperm was pelleted by centrifugation and the fixative was replaced with Hank's saline. Spermatozoa were spun onto an adhesive slide using a CytoSpin 4 (Thermo Fisher Scientific) at 800 rpm for 3 min. Slides were washed once in PBS, and sperm was permeabilized in 0.25% Tween in PBS for 30 min before blocking with 10% normal goat serum (Invitrogen) and 40 μg/ml BSA in PBST for at least 1 h. Slides were then incubated with mouse anti-zebrafish-Dcst2 antibody in blocking buffer (1:650) overnight at 4 °C in a humidified chamber. After several washes with PBST, slides were incubated with goat anti-mouse IgG Alexa Fluor 488 secondary antibody (1:380, Thermo Fisher Scientific) for 1 h, washed several times with PBST and finally once with PBS. After mounting using VECTASHIELD Antifade with DAPI (Vector Laboratories), sperm was imaged with an Axio Imager.Z2 microscope (Zeiss) using an oil immersion 100x/1.4 plan-apochromat objective. Widefield sperm images were processed for each genotype using Fiji by adjusting image brightness and contrast without clipping of intensity values.

**Co-IP.** Protein extracts [1 mg (TGC), 95~105 μg (sperm), and 200 μg (HEK293T)] were incubated with anti-HA antibody coated Dynabeads Protein G for immunoprecipitation (10009D, Thermo Fisher Scientific) for 1 h at 4 °C. After washing with a buffer [50 mM Tris-HCl (pH7.5), 150 mM NaCl, 0.1% Triton X-100, and 10% Glycerol], protein complexes were eluted with SDS sample buffer containing β-mercaptoethanol (for western blotting).

**HEK293T-oocyte binding assay.** Mouse Dcst1 ORF-3xFLAG, mouse Dcst2 ORF-3xHA, mouse Izumo1 ORF-1D4 with a Kozak sequence (gccgcc) and a rabbit polyadenylation [poly (A)] signal were inserted under the CAG promoter. These plasmids (0.67 μg/each, total 2 μg) were transfected into HEK293T cells using the calcium phosphate-DNA coprecipitation method[50]. After 2 days of transfection, these cells were resuspended in PBS containing 10 mM (ethylenedinitrilo)tetra-acetic acid. After centrifugation, the cells were washed with PBS, and then incubated with ZP-free eggs. After 30 min and then >6 h of incubation, the attached and fused cell numbers were counted under a fluorescence microscope (BZ-X700, Keyence) and an inverted microscope with relief phase contrast (IX73, Olympus). Proteins were extracted from the remaining HEK293T cells with a lysis buffer containing Triton-X 100 [50 mM NaCl, 10 mM Tris-HCl (pH 7.5), 1% (vol/vol) Triton-X 100 (Sigma–Aldrich)] containing 1% (vol/vol) protease inhibitor mixture, and then used for western blotting and co-IP.

**Fertility assessment of adult zebrafish.** The evening prior to mating, the fish assessed for fertility and a TLAB wild-type fish of the opposite sex were separated in breeding cages. The next morning, the fish were allowed to mate. Eggs were collected and kept at 28 °C in E3 medium (5 mM NaCl, 0.17 mM KCl, 0.33 mM CaCl₂, 0.33 mM MgSO₄, 10⁻⁵% Methylene Blue). The rate of fertilization was assessed approximately 3 h post-laying. By this time, fertilized embryos have developed to ~1000-cell stage embryos, while unfertilized eggs resemble one-cell stage embryos. Direct comparisons were made between siblings of different genotypes (wild-type, heterozygous mutant, homozygous mutant).

**Collection of zebrafish eggs and sperm.** Un-activated zebrafish eggs and sperm were collected following standard procedures[51]. The evening prior to sperm collection, male and female zebrafish were separated in breeding cages (one male and one female per cage).

To collect mature, un-activated eggs, female zebrafish were anesthetized using 0.1% w/v tricaine (25× stock solution in dH₂O, buffered to pH 7.0–7.5 with 1 M Tris pH 9.0). After being gently dried on a paper towel, the female was transferred to a dry petri dish, and eggs were carefully expelled from the female by applying mild pressure on the fish belly with a finger and stroking from anterior to posterior. The eggs were separated from the female using a small paintbrush, and the female was transferred back to the breeding cage filled with fish water for recovery.

To collect wild-type or mutant sperm, male zebrafish were anesthetized using 0.1% tricaine. After being gently dried on a paper towel, the male fish was placed belly-up in a slit in a damp sponge under a stereomicroscope with a light source

from above. Sperm were collected into a glass capillary by mild suction while gentle pressure was applied to the fish's belly. Sperm were stored in ice-cold Hank's saline (0.137 M NaCl, 5.4 mM KCl, 0.25 mM Na₂HPO₄, 1.3 mM CaCl₂, 1 mM MgSO₄, and 4.2 mM NaHCO₃). The male was transferred back to the breeding cage containing fish water for recovery. For western blot analysis, sperm from 3 males was sedimented at $800 \times g$ for 5 min. The supernatant was carefully replaced with 25 µL RIPA buffer [50 mM Tris-HCl (pH 7.5), 150 mM NaCl, 1 mM MgCl₂, 1% NP-40, 0.5% sodium deoxycholate, 1X complete protease inhibitor (Roche)] including 1% SDS and 1 U/µL benzonase (Merck). After 10 min of incubation at RT, the lysate was mixed and sonicated 3 times for 15 s of 0.5-s pulses at 80% amplitude (UP100H, Hielscher) interspersed by cooling on ice.

**Zebrafish sperm approach and binding assays**
*Imaging of zebrafish sperm approach.* Sperm were squeezed from 2–4 wild-type and mutant male fish and kept in 150 µL Hank's saline containing 0.5 µM MitoTracker Deep Red FM (Molecular Probes) for >10 min on ice. Un-activated, mature eggs were obtained by squeezing a wild-type female. To prevent activation, eggs were kept in sorting medium (Leibovitz's medium, 0.5% BSA, pH 9.0) at RT. The eggs were kept in place using a petri dish with cone-shaped agarose molds (1.5% agarose in sorting medium) filled with sorting medium. Imaging was performed with a LSM800 Examiner Z1 upright system (Zeiss) with a 20×/1.0 Plan-Apochromat water dipping objective. Before sperm addition, sorting media was removed and 1 mL of E3 medium was carefully added close to the egg. 5–10 µL of stained sperm was added as close to the egg as possible during image acquisition. The resulting time-lapse movies were analyzed using FIJI.

*Imaging and analysis of zebrafish sperm-egg binding.* Sperm was squeezed from 2 to 4 wild-type and mutant male fish and kept in 100 µL Hank's saline + 0.5 µM MitoTracker Deep Red FM on ice. Un-activated, mature eggs were squeezed from a wild-type female fish and activated by addition of E3 medium. After 10 min, 1–2 eggs were manually dechorionated using forceps and transferred to a cone-shaped imaging dish with E3 medium. After focusing on the egg plasma membrane, the objective was briefly lifted to add 2–10 µL of stained sperm (~200,000–250,000 sperm). Imaging was performed with a LSM800 Examiner Z1 upright system (Zeiss) using a 10x/0.3 Achroplan water dipping objective. Images were acquired until sperm were no longer motile (5 min). To analyze sperm-egg binding, stably-bound sperm were counted. Sperm were counted as bound when they remained in the same position for at least 1 min following a 90-s activation and approach time window. Data was plotted as the number of sperm bound per 100 µm of egg membrane for one minute.

**Statistics and reproducibility**. All values are shown as the mean ± SD of at least three independent experiments. Statistical analyses were performed using the two-tailed Student's *t* test (Figs. 4C, and S7B), Mann–Whitney *U* test (Figs. 5F and S7B), Steel–Dwass test (Figs. 2D, S6, and S8B), Kruskal–Wallis test with Dunn's multiple-comparisons test (Fig. 5A), and one-way ANOVA and multiple comparisons analysis (Fig. 5C) after examining the normal distribution and variance.

**Reporting summary**. Further information on research design is available in the Nature Research Reporting Summary linked to this article.

## Data availability

RNA-seq data reported here (zebrafish adult tissues) were deposited at the Gene Expression Omnibus (GEO) and are available under GEO acquisition number GSE171906. The newly generated plasmids will be available through Addgene [pClgn1.1-Dcst1-3xHA (#183540), pClgn1.1-Dcst2-3xHA (#183541), pCAG-Dcst1-3xFLAG (#183542), pCAG-Dcst2-3xHA (#183543), and pCAG-Izumo1-1D4 (#183544)]. The authors declare that the data that support the findings of this study are available from the corresponding authors upon request.

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

## Acknowledgements

We thank Natsuki Furuta, Eri Hosoyamada, Naoko Nagasawa, and the Biotechnology Research and Development (nonprofit organization) for excellent technical assistance; Alexander Schleiffer for phylogenetic analyses; Carina Pribitzer for RNA-Seq library preparation for zebrafish adult tissues; Mirjam Binner, Anna Kogan and the animal facility personnel from the IMP for help with genotyping and taking excellent care of zebrafish; Karin Aumayr and her team of the biooptics facility at the Vienna BioCenter (VBC) for support with microscopy; the Protein Technology Facility and the Next Generation Sequencing Facility at Vienna BioCenter Core Facilities (VBCF) for recombinant protein expression and zebrafish adult tissue RNA-Seq, respectively; the Max Perutz Labs Monoclonal Antibody Facility for generating anti-zebrafish Dcst2 antibodies; the entire Pauli lab for fruitful discussions, and Ms Ferheen Abbasi for critical reading of the manuscript. This work was supported by Ministry of Education, Culture, Sports, Science and Technology/Japan Society for the Promotion of Science KAKENHI (Grants-in-Aid for Scientific Research) Grants JP18K14612 and JP20H03172 to T.N., JP15H05573, JP16KK0180, JP20KK0155, JP21K19198, and JP21H02397 to Y.F., JP19J21619 to S.O., JP18K16735 to Y.L., and JP19H05750, and JP21H04753 to M.I.; Japan Agency for Medical Research and Development Grant JP21gm5010001 to M.I.; JST, PRESTO Grant JPMJPR2148 to T.N.; Takeda Science Foundation grants to T.N., Y.F. and M.I.; The Nakajima Foundation to T.N.; Mochida Memorial Foundation for Medical and Pharmaceutical Research grant to Y.F.; The Sumitomo Foundation Grant for Basic Science Research Projects to Y.F.; Senri Life Science Foundation grant to Y.F.; Intramural Research Fund (grants 21-2-6, 30-2-5 and 31-6-3) for Cardiovascular Diseases of National Cerebral and Cardiovascular Center to Y.F.; Eunice Kennedy Shriver National Institute of Child Health and Human Development Grants R01HD088412 and P01HD087157 to M.I.; and the Bill & Melinda Gates Foundation (grant INV-001902 to M.I.). Work in the Pauli lab has been supported by the FWF START program (Y 1031-B28 to A.P.), the HFSP Career Development Award (CDA00066/2015 to A.P.), a HFSP Young Investigator Award to A.P., EMBO-YIP funds to A.P., a Boehringer Ingelheim Fonds (BIF) PhD fellowship to A.B., a HFSP postdoctoral fellowship to V.E.D., and a DOC PhD student fellowship from the Austrian Academy of Sciences to K.R.G. The IMP receives institutional funding from Boehringer Ingelheim and the Austrian Research Promotion Agency (Headquarter grant FFG-852936).

## Author contributions

T.N., A.P. and M.I. conceptualized research; T.N., A.B., Y.F., K.R.G., C.E., V.E.D., S.O., K.P., Y.L., S.B and M.K. performed research; T.N., A.B., Y.F., K.R.G., C.E., V.E.D., K.P., S.B. and M.I. analyzed data; L.E.C.Q. performed analysis of RNA-Seq data; and T.N., A.B., K.R.G., V.E.D., K.P., A.P. and M.I. wrote the paper.

## Competing interests

The authors declare no competing interests.
