## [Peer Review File · Communications Biology]

Reviewers' comments:

Reviewer #1 (Remarks to the Author):

In this paper, the authors investigate the role of two multipass membrane proteins in mice and fish; they generate single and double knockouts and find that the proteins are required for fertilization. In particular, the authors come to the conclusion that DCST1 and DCST2 are essential for the binding of sperm and eggs in zebrafish, while in mouse they are required for sperm-egg fusion. This work confirms the same finding in mice that was recently reported by another group (PMID: 33871360) and extends the study to an evolutionarily distant vertebrate species.

The title is not consistent with the authors' conclusions; therefore, it should be changed accordingly.

Major points:

- Line 131: the expression levels of Dcst1 and Dcst2 in the double heterozygous males is not shown, even if used to infer that they are sufficient to maintain male fertility
- Figure 2 panels B and E, if the data were reported as the number of acrosome-reacted sperm bound to the oolemma, would the difference be significant? To better assess the binding ability of sperm to the oolemma, only acrosome-reacted sperm should be counted. Acrosome-intact sperm are unable to fuse with the egg and could be a confounding factor. Moreover, given that DCST2 is required for sperm-egg binding in zebrafish it is relevant to establish its precise role in mice as well.
- Figure 3B: are the Dcst1 expression levels restored in Dcst2d25/d25; Tg?
- Line 231: it would be interesting to know if, in zebrafish, the expression of Dcst1 is reduced in the Dcst2 KO and vice versa, as it happens in mice.
- Inoue et al (PMID: 33871360) have observed that SPACA6 disappears from the mature spermatozoa of their DCST1/2 KO. Have the authors checked if they find the same? Is SPACA6 present in the Dcst1d1/d1 and Dcst2 d25/d25 sperm?

Minor points:

Figure S1A: the tree scale is unclear

Line 375: which expression system was used to produce the recombinant zebrafish Dcst2?

Figure 4A: why are the expression levels of all proteins lower in the co-transfected cells? Could this affect the binding assay (Fig 4B and 4C?)

Reviewer #2 (Remarks to the Author):

The article by Noda et al investigates the potential role of 2 genes, DCST1 and DCST2, in the gamete interaction process. The authors suspected a role for these genes in sperm-egg fusion, given their belonging to the DC-STAMP family of genes involved in cell-cell fusion, from yeast to mammals.

This manuscript describes the KO of each of these 2 genes, separately and together in a mouse model. It demonstrates the essential aspect of these genes in the process, as males KO for DCST2 are sterile and those for DCST1 exhibit a severe infertility. The authors detail the phenotypes of these mice. They have also performed a similar KO in the zebrafish model and obtained comparable results.

The paper is well written and complete. It only suffers from the fact that a very similar article has already been published by Inoue et al in eLife, in April 2021. There are no major concerns to report. Only some rare remarks that could be included in the final version.

Questions and remarks :

In Fig 1B. :

Knowing that the KO of Tmem95 is sterile, the apparent absence of TMEM95 expression is intriguing, the authors can at least cite their previous work discussing the low level of expression or discuss this point.

L114-115 :

“tandemly arranged” : I would have said “inversely duplicated or in mirror” but I think “tandemly” means duplicated in the same orientation. Could the authors check this point?

DCST1 labelling :

It is said that HA-tagged DCST1 could rarely be observed in spermatozoa, maybe the authors could show the image. The authors could also discuss this weak level of expression.

Fig 2 B and E:

What are the conditions of this experiment ? How long is the insemination before counting the bound spermatozoa ? WT oocytes in the presence of control sperm should be fertilized?

Discussion :

299 The function of DCST1/2 in the sperm-egg fusion process differs between mice and

300 fish: mouse DCST1/2 are required for the fusion process after sperm-egg binding

301 (Figure 2), while zebrafish Dcst1/2 are required for sperm-egg binding (Figure 5).

There is no evidence that DCST1/2 are directly involved in the fusion step. As for Izumo1, the adhesion remains after their deletion, however IZUMO1 is indeed a proven adhesion protein.

302 Given the diversity of the fertilization process across the animal kingdom, it may be that

303 while DCST proteins are highly conserved...

Why do the authors claim that the level of conservation is high? Related to what?

Typing mistakes:

- Suppl L32 : pronuclear should be replaced by pronucleus
- Suppl Fig S3 B : gRNA#1 should probably be #3. Please check.
- Maybe detail at least once in the text DCSTAMP, OCSTAMP and DCST.

Reviewer #3 (Remarks to the Author):

In the manuscript, "Sperm membrane proteins DCST1 and DCST2 are required for the sperm-egg fusion process in mice and fish" by Ikawa et al., the authors described the importance of DCST1 and DCST2 in fusion of sperm with oolemma in vertebrate species. This is a second report in mice and first of its kind in fish. The findings of the manuscript may generate wider interest among the reproductive biologists.

The well designed methodology and clearly drawn interpretation are strength of this paper.

General minor comments:

1. The consistency in writing sperm/spermatozoa has to be maintained
2. line 135: DCST2 is indispensable. Agreed. clarity is missing in Line 138, wherein it was mentioned that DCST mutant mice rarely fertilise eggs?. Also, in the subsequent paragraph/results it was reported that sperm from KO (DCST1 and DCST2 genes) males did fuse with eggs, without embryo formation.

3. Materials:

Though the headings are specific, the sub-headings can be clubbed together to provide complete picture. For example, instead of separate heading for antibodies (line 364), this can be clubbed with immunocytochemistry (line 434).

In the western and immunocytochemistry loading control and negative control, respectively may be mentioned clearly.

Line 333: RT-PCR: not clear whether this was realtime PCR or not? Whether MIQE Guidelines were followed while setting up the experiment?

Reviewer #1:

In this paper, the authors investigate the role of two multipass membrane proteins in mice and fish; they generate single and double knockouts and find that the proteins are required for fertilization. In particular, the authors come to the conclusion that DCST1 and DCST2 are essential for the binding of sperm and eggs in zebrafish, while in mouse they are required for sperm-egg fusion. This work confirms the same finding in mice that was recently reported by another group (PMID: 33871360) and extends the study to an evolutionarily distant vertebrate species.

Comment #1: The title is not consistent with the authors' conclusions; therefore, it should be changed accordingly.

Response: As suggested by the reviewer, we changed the title to “sperm-egg interaction” (Page 1 line 2).

Comment #2: Major points:

- Line 131: the expression levels of *Dcst1* and *Dcst2* in the double heterozygous males is not shown, even if used to infer that they are sufficient to maintain male fertility

Response: We collected the testicular cDNA from double heterozygous mice (*Dcst1*^{d1/wt} and *Dcst2*^{d25/wt}), and then performed quantitative PCR (qPCR). As shown in the following Figure, the expression levels of *Dcst1* and *Dcst2* mRNAs in dHZ testes is comparable to the control. We further examined the quantitative expression level of *Dcst1/2* using more mutants because the expression of *Dcst1* seemed to decrease in a *Dcst2*^{d25/d25} testis (**Figure S3C**). However, we could not detect a significant difference in *Dcst1/2* mRNA levels between *Dcst1* KO, *Dcst2* KO as shown in the following Figure, and control testes indicating that deletion of one paralog does not affect the transcription of the other. Thus, we removed the dHZ data, and made a new **Figure S4**.

We revised the text in the MS as shown below (Page 3, line 118-154) and added the methods of qPCR to the Supplemental information:

“Using CRISPR/Cas9-mediated mutagenesis, we generated *Dcst2* mutant mice lacking 7,223 bp (*Dcst2*^{del/del}), which deleted most of the *Dcst2* open reading frame (ORF) (Figure S2A-D). As shown in Figure S2A, *Dcst1* and *Dcst2* are adjacent genes in a head-to-head arrangement such that parts of their 5' genomic regions overlap. Because deletion of *Dcst2* could in principle affect *Dcst1* transcription, we also generated indel mice, *Dcst1*^{d1/d1} and *Dcst2*^{d25/d25} (Figure S3A and B). RNA isolation from mutant testes followed by cDNA sequencing revealed that *Dcst1*^{d1/d1} has a 1-bp deletion in exon 1, and *Dcst2*^{d25/d25} has a 25-bp deletion in exon 4 (Figure S3C and D). Both deletions result in frameshift mutations leading to premature stop codons (Figure S3E).

Dcst1^{d1/d1}, *Dcst2*^{d25/d25}, and *Dcst2*^{del/del} male mice successfully mated with female mice. However, crosses between *Dcst2*^{d25/d25} and *Dcst2*^{del/del} males and wild-type females did not result in any offspring, and crosses with *Dcst1*^{d1/d1} males were only rarely giving rise to pups {pups/plug: 9.01 ± 2.77 [control (Ctrl), 19 plugs], 0.22 ± 0.19 [*Dcst1*^{d1/d1}, 17 plugs], 0 [*Dcst2*^{d25/d25}, 42 plugs], 0 [*Dcst2*^{del/del}, 24 plugs]}, indicating that *Dcst1* mutant males are almost and *Dcst2* males are completely sterile (Figure 1C). Together with our finding that the levels of *Dcst1* mRNA were similar between wild type and the two different *Dcst2* mutant testes (Figure S2D, S3C, and S4), this suggests that (1) the sterility of *Dcst2* mutants was caused by the loss of *Dcst2* expression and not by a concomitant decrease in *Dcst1* expression; and (2) both DCST1 and DCST2 are required for fertilization. Hereafter, we used *Dcst1*^{d1/d1} and *Dcst2*^{d25/d25} male mice for all experiments unless otherwise specified.”

Comment #3: Figure 2 panels B and E, if the data were reported as the number of acrosome-reacted sperm bound to the oolemma, would the difference be significant? To better assess the binding ability of sperm to the oolemma, only acrosome-reacted sperm should be counted. Acrosome-intact sperm are unable to fuse with the egg and could be a confounding factor. Moreover, given that DCST2 is required for sperm-egg binding in zebrafish it is relevant to establish its precise role in mice as well.

Response: Thanks for the reviewer's constructive suggestions. The number of acrosome-reacted (AR) sperm of *Dcst1* and *Dcst2* KO mice that are bound to the oolemma is significantly higher than for control sperm (see the following figure), indicating that AR sperm of *Dcst1/2* KO mice can bind to oocytes. We analyzed the binding and fusion ability 30 minutes after sperm addition – a time-point at which some control sperm had

already fused with oocytes (**Figure 2E-F**). The observed increased binding of mutant sperm compared to wild-type sperm to the oolemma is likely a consequence of the inability of mutant sperm to fertilize the egg since successful fertilization (with wild-type sperm) induces changes to the oolemma that prevent further sperm from binding and entering. Our data is therefore consistent with the idea that successful fertilization leads to the decreased number of AR sperm bound to oolemma in control mice. In fact, Bianchi et al. (Nature, 2014) showed that JUNO, the IZUMO1 receptor on the egg, was barely detectable on zona-free oocytes 30 to 40 minutes after fertilization, leading to the membrane block to polyspermy (Bianchi E et al., Nature 2014).

Hence, we edited Figure 2 and the MS as shown below (Pages 4, lines 173-212):

“To examine the binding and fusion ability of *Dcst1^{d1/d1}* and *Dcst2^{d25/d25}* mutant sperm, we incubated mutant sperm with ZP-free eggs. Both types of mutant sperm could bind to the oolemma [5.72 ± 1.97 (Ctrl, 113 eggs), 7.64 ± 4.68 (*Dcst1^{d1/d1}*, 89 eggs), 7.63 ± 3.45 (*Dcst2^{d25/d25}*, 89 eggs)] (**Figure 2A**). Because binding is not defective in mutant sperm, we confirmed that IZUMO1, a key factor in this process, was expressed normally in testicular germ cells (TGCs) and sperm of *Dcst1^{d1/d1}* and *Dcst2^{d25/d25}* males (**Figure 2B**). Indeed, we found that the level of IZUMO1 in mutant sperm was comparable to the control (**Figure 2B**). Moreover, the oolemma-bound sperm of *Dcst1^{d1/d1}* and *Dcst2^{d25/d25}* males underwent the acrosome reaction (AR) normally as determined by live-cell staining with IZUMO1 antibody (**Figure 2C**). The number of acrosome-reacted mutant sperm bound to the oolemma was significantly higher than the number of control sperm [3.27 ± 2.31 (Ctrl), 7.34 ± 5.09 (*Dcst1^{d1/d1}*), 4.74 ± 2.93 (*Dcst2^{d25/d25}*)] (**Figure 2D**).

To assess the ability of mutant sperm to fuse with the oolemma, sperm were incubated with Hoechst 33342-preloaded ZP-free eggs. In experiments with control heterozygous sperm, Hoechst 33342 fluorescence signal was translocated into sperm heads (**Figure 2E**, arrow), indicating that these sperm fused with the egg membrane. However, Hoechst 33342 signal was rarely detected in *Dcst1* KO and not detected in *Dcst2* KO sperm bound to the oolemma [fused sperm/egg: 1.52 ± 0.35 (Ctrl, 113 eggs), 0.04 ± 0.05 (*Dcst1*^{d1/d1}, 73 eggs), 0 (*Dcst2*^{d25/d25}, 73 eggs)] (**Figure 2E and F**). These results indicate that control heterozygous sperm can fuse with eggs but *Dcst2* KO and *Dcst1* KO sperm are impaired at the step of fusion (**Figure 2E**). The fusion defect is in agreement with the increased number of sperm bound to the oolemma due to the absence of the membrane block of polyspermy that is normally triggered by fertilization¹. Thus, while acrosome-reacted sperm of *Dcst1* and *Dcst2* KO mice can bind to eggs, they are defective at fusing with the oolemma: KO males of *Dcst1* or *Dcst2* causes a strong impairment or complete loss of sperm-egg fusion, respectively.”

Comment #4: Figure 3B: are the Dcst1 expression levels restored in Dcst2d25/d25; Tg?

Response: We do not have antibodies against mouse DCST1/2. Thus, we examined the expression level of *Dcst1* mRNA in *Dcst2* KO Tg mice using qPCR. The expression level is comparable to the control (also see comment #2). We added **Figure S4** containing this result in the revised MS (Pages 3-4 lines 138-142). While this is an interesting and valid question, we are therefore unfortunately not able to analyze the protein expression level of mouse DCST1 in *Dcst2* mutants. However, given our experiments in zebrafish that reveal a clear interdependency of the protein levels of Dcst1 and Dcst2 (new **Figure 5C**), we can by analogy infer that the protein levels of DCST1 will likely be strongly reduced or absent in *Dcst2* KOs despite the presence of *Dcst1* mRNA (see **Figure S4**) but will be increased again in the presence of a *Dcst2* transgene.

Comment #5: - Line 231: it would be interesting to know if, in zebrafish, the expression of Dcst1 is reduced in the Dcst2 KO and vice versa, as it happens in mice.

Response: Thank you for bringing up this interesting point. While we previously only had an antibody that detected endogenous zebrafish Dcst2, we have in the meanwhile obtained an antibody against endogenous zebrafish Dcst1 that is specific against Dcst1. In addition, we optimized the protocol for immunoblotting for both Dcst1 and Dcst2 antibodies (biggest differences: semi-native conditions as published in Zhang et al., Nat Commun, 12, 4380, 2021: reduced SDS and DTT, and no boiling of the samples but

incubation at 42°C instead). Using this optimized protocol, we indeed observed that we cannot detect Dcst1 protein in the absence of Dcst2, and vice versa cannot detect Dcst2 protein in the absence of Dcst1. We have included this new set of data in our main Figure 5 (**Figure 5C**). For completeness, we have also exchanged the brightfield (BF) and immunofluorescence images (previous Figure 5 panels B and C) to show BF and immunofluorescence images of Dcst2 protein localization in WT, *dcst1*^{-/-}, *dcst2*^{-/-} and *dcst1/2*^{-/-} sperm (new **Figure 5B**). (Previously, we only showed images for the WT and *dcst2*^{-/-}). Please note that the Dcst1 antibody does not work for immunofluorescence as we were unable to detect Dcst1 protein in WT sperm (data not shown).

We have updated the relevant parts in the manuscript (text, Figure, legends, and methods): “To understand what causes the fertility defect, we first determined whether sperm were produced in mutant males. *Dcst1*^{-/-}, *dcst2*^{-/-}, and *dcst1/2*^{-/-} males showed normal mating behavior and produced morphologically normal sperm (DIC images in **Figure 5B**), indicating that zebrafish Dcst1/2 are not crucial for spermatogenesis. To detect Dcst1 and Dcst2 proteins, we produced antibodies against the C-terminal RING finger domains of zebrafish Dcst1 and Dcst2. Each antibody was specific against its cognate target antigen as determined by western blotting of wild-type and KO sperm lysates (**Figure 5C**), and Dcst2 was detected by immunofluorescence staining of zebrafish embryos overexpressing *dcst2(RING)-superfolder GFP (sfGFP)* mRNA (**Figure S8D**). Interestingly, *dcst1*^{-/-} and *dcst2*^{-/-} sperm were not only lacking Dcst1 or Dcst2 protein, respectively, but were lacking both Dcst proteins (**Figure 5C**). This suggests that Dcst1/2 protein stability requires the presence of both proteins, which is consistent with mouse DCST1/2 forming a protein complex (**Figure 3E**). To examine where Dcst2 is localized in zebrafish sperm, which lacks an acrosome, we performed immunofluorescence against Dcst2. Wild-type sperm was strongly stained at the periphery of the head in foci and the mid-piece (**Figure 5B**). Staining of the mid-piece and occasionally the tail region was also detected in *dcst2* KO sperm, suggesting that this signal was unrelated to Dcst2. Dcst2 foci are markedly reduced in *dcst1*^{-/-} sperm, which is consistent with the observed interdependence of Dcst1/2 in sperm lysates.”

Updated legend of Figure 5:

B) Dcst2 localizes to the periphery of the sperm head. Immunofluorescent and differential interference contrast (DIC) images of wild-type and mutant zebrafish sperm. Sperm were stained with DAPI (blue) to visualize nuclei and an antibody against the RING-finger domain of zebrafish Dcst2. Dcst2 localizes to distinct foci around the sperm

head (green arrowheads). Dcst2 foci are reduced (*dcst1*^{-/-}) or not detectable (*dcst2*^{-/-}; *dcst1/2*^{-/-}) in mutant sperm, while overall sperm morphology in DIC images appears normal for all genotypes. Autofluorescence of the sperm mid-piece appears as a uniform signal in the Dcst2 channel (orange arrowhead) in all genotypes. Scale bar: 5 μm. Boxed inset shows an individual representative sperm head for each genotype (scale bar: 2 μm).

C) Dcst1 and Dcst2 are absent in mutant sperm. Exemplary immunoblot of sperm samples of wild type and mutant genotypes probed with antibodies against zebrafish Dcst1 and Dcst2. Tubulin protein levels of the same blot are shown as loading control. For Dcst2, an unspecific band (*) is detected in all genotypes at ~250 kDa. Quantification of Dcst2 levels of wild-type and mutant sperm based on four independent immunoblots. Values were normalized to tubulin and then to wild-type levels. Statistical significance was calculated using one-way ANOVA and multiple comparisons analysis: ****p < 0.0001.

Updated methods part (added the generation of the Dcst1 antibody (see also Comment #8) and the optimized Western and IF protocol). The mouse and zebrafish protocols are now separated for increased accessibility:

Western blotting.

Mouse samples were mixed with sample buffer containing β-mercaptoethanol⁴⁰, and boiled at 98°C for 5 minutes prior to SDS PAGE. After protein transfer, the polyvinylidene difluoride (PVDF) membrane was treated with Tris-buffered saline (TBS)-0.1% Tween20 (Nacalai Tesque) containing 10% skim milk (Becton Dickinson and Company) for 1 hour, followed by the primary antibody [IZUMO1, SLC2A3, HA, and FLAG (1:1,000), 1D4 (1:5,000)] for 3 hours or overnight. After washing with TBST, the membrane was treated with secondary antibodies (1:1,000). The HRP activity was visualized with ECL prime (BioRad) and Chemi-Lumi One Ultra (Nacalai Tesque). Then, the total amount of proteins on the membrane was visualized with Coomassie Brilliant Blue (CBB) (Nacalai Tesque).

Zebrafish sperm from 3-6 males was sedimented at 3000 rpm for 3.5 minutes. The supernatant was replaced with modified RIPA buffer [50 mM Tris-HCl (pH 7.5), 150 mM NaCl, 1 mM MgCl₂, 1% NP-40, 0.5% sodium deoxycholate, 1% SDS, 1x complete protease inhibitor (Roche)]. After preparation of all samples, 1 U/μl benzonase (Merck) was added, and samples were incubated for 30 minutes at room temperature. Samples were then mixed with semi-native SDS-PAGE sample buffer [100 mM Tris-HCl (pH 6.8), 8% glycerol, 0.1 mg/ml bromophenol blue, 2% SDS, 10 mM DTT]⁵⁰ and heated to 42°C

for 5 minutes, or alternatively, mixed with 4x Laemmli containing β -mercaptoethanol and boiled at 95°C for 5 minutes. After SDS-PAGE, samples were wet-transferred onto a nitrocellulose membrane. Total protein was visualized by Ponceau staining before blocking with 5% milk powder in 0.1% Tween in 1x TBS (TBST). Membranes were incubated in primary antibody overnight at 4°C, then washed with TBST before HRP-conjugated secondary antibody [1:10.000 (115-036-062, Dianova)] incubation for 1 hour. Membranes were washed several times in TBST before HRP activity was visualized using Clarity Western ECL Substrate (BioRad) on a ChemiDoc (BioRad). After detection, membranes were stripped using Restore Western Blot Stripping Buffer (Thermo Fisher Scientific) before re-blocking and proceeding with another primary antibody incubation. Primary antibodies: mouse anti-zebrafish-Dcst1 (1:25); mouse anti-zebrafish-Dcst2 (1:500); mouse anti-alpha-Tubulin [1:20.000 (T6074, Merck)]. To assess relative Dcst2 protein amounts, average intensities of Dcst2-specific bands were quantified in Fiji on 4 independent immunoblots for each genotype relative to Tubulin levels, which was used as loading control. Values were then normalized to the levels of wild-type sperm.

Comment #6:- Inoue et al (PMID: 33871360) have observed that SPACA6 disappears from the mature spermatozoa of their DCST1/2 KO. Have the authors checked if they find the same? Is SPACA6 present in the Dcst1 d1/d1 and Dcst2 d25/d25 sperm?

Response: As shown in the image below, SPACA6 (the predicted size is about 37 kDa) is strongly diminished in *Dcst1* KO and *Spaca6* KO sperm compared to the control. However, we could see a faint signal even in *Spaca6* KO sperm. These signals may be background and non-specific, but we need to generate a new antibody against SPACA6 in the future to obtain a clearer results. Thus, we would like to show the following data to the reviewers only.

Hence, as suggested by the reviewer, we cited the previous paper (Inoue et al., eLife, 2021), and discuss it in the discussion as shown below (Page 8, lines 331-340):

“Having established the essential role of DCST1/2 for male fertility, questions arise concerning the molecular processes in which they are involved and how they contribute to fertilization. We detected mouse DCST2 at the equatorial segment of acrosome-reacted sperm and zebrafish Dcst2 at the periphery of the sperm head. Since DCST1/2 are TM proteins, these localization patterns suggest that part of the proteins are exposed on the sperm surface. Given our result that DCST1/2 form a complex (**Figure 3E**) and are interdependent of each other (**Figure 5C**), we speculate that a DCST1/2 complex in the sperm membrane helps organize the presentation of other fusion-related sperm proteins. The recently reported absence of SPACA6 in *Izumol* or *Dcst1/2* KO sperm supports this idea³³.”

Minor points:

Comment #7: Figure S1A: the tree scale is unclear

Response: Thank you for pointing out that the tree scale in Figure S1A had not been explained properly – we apologize for this oversight. The branch lengths represent the expected number of substitutions per site. The label of the scale bar was corrected, and the figure legend was complemented.

We have added the following explanation to the Figure legend:

“The branch lengths represent the number of substitutions per site.”

Comment #8: Line 375: which expression system was used to produce the recombinant zebrafish Dcst2?

Response: We have updated the methods part to include the expression system used to generate the Dcst2 (and also Dcst1) recombinant protein fragments for antibody generation. Recombinant proteins were expressed in *E. coli* BL21(DE3).

The new methods section has now the following wording:

“To generate mouse monoclonal antibodies against zebrafish Dcst1 and Dcst2, recombinant zebrafish Dcst1 (amino acids 590-675) and Dcst2 (amino acids 574-709) proteins were expressed in *E. coli* BL21(DE3) and purified by the VBCF Protein Technologies Facility. Each recombinant protein was injected into 3 mice, and monoclonal antibodies were generated by the Max Perutz Labs Monoclonal Antibody Facility according to standard procedures.”

Comment #9: Figure 4A: why are the expression levels of all proteins lower in the co-transfected cells? Could this affect the binding assay (Fig 4B and 4C?)

Response: As we used the CAG promoter for all plasmids, the decrease in the expression level may be caused by the competition of transcription factors to bind to the multiple constructs with the same promoter, leading to the decreased number of HEK293T cells expressing *Dcst1/2* and *Izumo1* bound to eggs (Figure 4B and 4C). However, we note that the cells can bind to eggs even if the expression level is lower.

Reviewer #2:

Comment #10: The article by Noda et al investigates the potential role of 2 genes, DCST1 and DCST2, in the gamete interaction process. The authors suspected a role for these genes in sperm-egg fusion, given their belonging to the DC-STAMP family of genes involved in cell-cell fusion, from yeast to mammals.

This manuscript describes the KO of each of these 2 genes, separately and together in a mouse model. It demonstrates the essential aspect of these genes in the process, as males KO for DCST2 are sterile and those for DCST1 exhibit a severe infertility. The authors detail the phenotypes of these mice. They have also performed a similar KO in the zebrafish model and obtained comparable results.

The paper is well written and complete. It only suffers from the fact that a very similar article has already been published by Inoue et al in eLife, in April 2021. There are no major concerns to report. Only some rare remarks that could be included in the final version.

Response: We thank the reviewer for the kind words and constructive remarks. We would like to note that we have independently obtained the results presented here, as can be seen by the date of our bioRxiv pre-print publication (v1 version was posted on 18.Apr.2021, <https://www.biorxiv.org/content/10.1101/2021.04.18.440256v2.article-info>) which was before the Inoue et al study came out in eLife (published 19. Apr. 2021). Moreover, Inoue et al only analyzed the function of DCST1/2 in mice, not in zebrafish – so the zebrafish data we present here is entirely novel. In addition, we find a phenotypic difference in the sperm-egg interaction between *Dcst1/2* KO mice and zebrafish: mouse DCST1/2 mutants can still bind to the egg and have a defect in sperm-egg fusion, while zebrafish *Dcst1/2* are necessary for sperm-egg binding. We therefore present important novel findings that confirm and significantly expand the findings by Inoue et al. We addressed the reviewer's comments as detailed below.

Questions and remarks:

Comment #11: In Fig 1B. :

Knowing that the KO of *Tmem95* is sterile, the apparent absence of *TMEM95* expression is intriguing, the authors can at least cite their previous work discussing the low level of expression or discuss this point.

Response:

We thank the reviewer for pointing out the seemingly absent expression levels of *Tmem95* in previous **Figure 1B**. In our data set we indeed do not detect higher levels of *Tmem95* in the time-course for unknown reasons. Since we do not refer to *Tmem95* further in our paper, we have decided to remove *Tmem95* expression from **Figure 1B** and also removed the reference to *Tmem95* from page 3, line 116 to avoid confusion.

Comment #12: L114-115 :

“tandemly arranged” : I would have said “inversely duplicated or in mirror” but I think “tandemly” means duplicated in the same orientation. Could the authors check this point?

Response: Thanks for pointing out the mistake. We revised the phrasing (Page 3, lines 122-124):

As shown in **Figure S2A**, *Dcst1* and *Dcst2* are adjacent genes in a head-to-head arrangement such that parts of their 5' genomic regions overlap.

Comment #13: DCST1 labelling :

It is said that HA-tagged DCST1 could rarely be observed in spermatozoa, maybe the authors could show the image. The authors could also discuss this weak level of expression.

Response: As shown below, HA-tagged DCST1 was barely detected in sperm. We added the results in **Figure S7C** and revised the relevant results section and Figure legend.

We revised the MS as shown below (Pages 5-6, lines 222-240).

“Although both transgenes code for a C-terminal HA-tag and could rescue *Dcst1/2* KO, only very low levels of HA-tagged DCST1 could be detected in sperm samples (**Figure S7C**), suggesting that only low levels of DCST1 are required for fertilization. HA-tagged DCST2 was detected in TGCs and sperm at the expected size for the full-length protein (**Figure 3B and S7C**, arrowheads), though both DCST1 and DCST2 proteins appear to be subject to post-translational processing or protein degradation.

To reveal the localization of DCST2 in sperm, we performed immunocytochemistry with an antibody detecting the HA epitope and peanut agglutinin (PNA) as a marker for the sperm acrosome reaction. As shown in **Figure 3C**, PNA in the anterior acrosome was translocated to the equatorial segment after the acrosome reaction as shown previously²⁹. HA-tagged DCST2 was detected within the anterior acrosome of acrosome-intact sperm, and then translocated to the equatorial segment after the acrosome reaction (**Figure 3C**), mirroring the relocalization of IZUMO1 upon the acrosome reaction³⁰. Fluorescence in the sperm tail was observed in both control wild-type and *Dcst2-HA* Tg sperm, indicating that this signal in the tail was non-specific.”

Comment #14: Fig 2 B and E:

What are the conditions of this experiment ? How long is the insemination before counting the bound spermatozoa ? WT oocytes in the presence of control sperm should be fertilized?

Response: We apologize for the confusion. To make the relevant method section more accessible (Page 10, line 456), we changed the section title to “Mouse sperm-egg binding and fusion assay”. Because we examined the sperm binding and fusion ability after 30 minutes of incubation, some control sperm already fused with oocytes (**Figure 2E-F**). As reviewer #1 suggested to count acrosome-reacted sperm bound to oolemma (see comment #3), we revised this part as shown below (Page 4, lines 173-205):

“To examine the binding and fusion ability of *Dcst1^{d1/d1}* and *Dcst2^{d25/d25}* mutant sperm, we incubated mutant sperm with ZP-free eggs. Both types of mutant sperm could bind to the oolemma [5.72 ± 1.97 (Ctrl, 113 eggs), 7.64 ± 4.68 (*Dcst1^{d1/d1}*, 89 eggs), 7.63 ± 3.45 (*Dcst2^{d25/d25}*, 89 eggs)] (**Figure 2A**). Because binding is not defective in mutant sperm, we confirmed that IZUMO1, a key factor in this process, was expressed normally in testicular germ cells (TGCs) and sperm of *Dcst1^{d1/d1}* and *Dcst2^{d25/d25}* males (**Figure 2B**). Indeed, we found that the level of IZUMO1 in mutant sperm was comparable to the

control (**Figure 2B**). Moreover, the oolemma-bound sperm of *Dcst1*^{d1/d1} and *Dcst2*^{d25/d25} males underwent the acrosome reaction (AR) normally as determined by live-cell staining with IZUMO1 antibody (**Figure 2C**). The number of acrosome-reacted mutant sperm bound to the oolemma was significantly higher than the number of control sperm [3.27 ± 2.31 (Ctrl), 7.34 ± 5.09 (*Dcst1*^{d1/d1}), 4.74 ± 2.93 (*Dcst2*^{d25/d25})] (**Figure 2D**).

To assess the ability of mutant sperm to fuse with the oolemma, sperm were incubated with Hoechst 33342-preloaded ZP-free eggs. In experiments with control heterozygous sperm, Hoechst 33342 fluorescence signal was translocated into sperm heads (**Figure 2E**, arrow), indicating that these sperm fused with the egg membrane. However, Hoechst 33342 signal was rarely detected in *Dcst1* KO and not detected in *Dcst2* KO sperm bound to the oolemma [fused sperm/egg: 1.52 ± 0.35 (Ctrl, 113 eggs), 0.04 ± 0.05 (*Dcst1*^{d1/d1}, 73 eggs), 0 (*Dcst2*^{d25/d25}, 73 eggs)] (**Figure 2E and F**). These results indicate that control heterozygous sperm can fuse with eggs but *Dcst2* KO and *Dcst1* KO sperm are impaired at the step of fusion (**Figure 2E**). The fusion defect is in agreement with the increased number of sperm bound to the oolemma due to the absence of the membrane block of polyspermy that is normally triggered by fertilization¹. Thus, while acrosome-reacted sperm of *Dcst1* and *Dcst2* KO mice can bind to eggs, they are defective at fusing with the oolemma: KO males of *Dcst1* or *Dcst2* causes a strong impairment or complete loss of sperm-egg fusion, respectively.”

Comment #15: Discussion :

The function of DCST1/2 in the sperm-egg fusion process differs between mice and fish: mouse DCST1/2 are required for the fusion process after sperm-egg binding (Figure 2), while zebrafish Dcst1/2 are required for sperm-egg binding (Figure 5).

There is no evidence that DCST1/2 are directly involved in the fusion step. As for Izumo1, the adhesion remains after their deletion, however IZUMO1 is indeed a proven adhesion protein.

Response: We thank the reviewer for this comment, and agree with the assessment. Throughout our manuscript, we have been trying to be careful not to overstate our conclusion, and therefore phrased our conclusion to describe the loss of function phenotypes which suggests at which step the protein is necessary. From our phenotypic analysis, it is clear that in mice, Dcst1/2 KO sperm can still bind to the oolemma and fail to complete fertilization at the step of sperm-egg membrane fusion, while in zebrafish they already fail to bind stably to the oolemma. We have revised these sentences in the discussion to make this point more clear (Page 8 lines 350-358):

“The role of DCST1/2 during sperm-egg interaction differs between mice and fish. Specifically, loss of mouse DCST1/2 led to a defect in sperm-egg fusion (**Figure 2**), suggesting that DCST1/2 may directly or indirectly regulate membrane fusion via an IZUMO1-independent pathway or act as fusion mediators downstream of the interaction between IZUMO1 and JUNO. Interestingly, zebrafish *Dcst1/2* are required for sperm-egg binding (**Figure 5**).”

Comment #16: Given the diversity of the fertilization process across the animal kingdom, it may be that while DCST proteins are highly conserved...

Why do the authors claim that the level of conservation is high? Related to what?

Response: We apologize for the imprecision. We are referring to DCST1/2 being widely conserved among bilaterians as discussed in the first paragraph of the discussion. DCST1/2 are the only vertebrate proteins involved in sperm-egg interaction that are currently known to be conserved outside of vertebrates since they have clear homologs for example in *C. elegans* (Spe-42/Spe-49) and *Drosophila* (Sneaky). This has been further documented in the literature [Binner et al, 2021 (bioRxiv, <https://www.biorxiv.org/content/10.1101/2021.11.19.469324v1>) – Figure panel 1D showing phylogenetic tree that reveals that *Dcst1/2* are conserved outside of vertebrates while *Izumol* and *Spaca6* are only conserved in vertebrates); Deneke and Pauli, *Annu Rev Cell Dev Biol* 37, 391-414, 2021; Vance and Lee, *Curr Biol* 30, R750-R754, 2020; Fujihara et al, *Proc Natl Acad Sci USA*, 118, 2021]. We therefore revised the discussion and cite the relevant literature (Page 8 lines 319-329, 358-361):

“Here, we demonstrate that the testis-enriched proteins DCST1/2 are necessary for male fertility in mice and fish. These results agree with the recently reported independent findings in mice³³ and the known requirements of *Drosophila* Sneaky and *C. elegans* SPE-42/49 in sperm for successful reproduction²¹⁻²⁵, indicating that the physiological function of DCST1/2 is widely conserved among bilaterians. The conservation of DCST1/2 outside of vertebrates is exceptional not only because fertilization-relevant genes are rapidly evolving^{34,35} but also because other known vertebrate proteins involved in sperm-egg interaction have orthologs only in select vertebrate genera³⁶⁻³⁹. Contrary to vertebrates, which have both DCST1/2 and the related DCSTAMP/OCSTAMP proteins (**Figure S1A**), invertebrates have solely DCST1/2.”

“Given the diversity of the fertilization process across the animal kingdom, it may be that while DCST proteins are widely conserved^{36,37}, they may have evolved different roles to

fit into the specific context of fertilization for a given species or genus.”

Comment #17: Typing mistakes:

- Suppl L32 : pronuclear should be replaced by pronucleus
- Suppl Fig S3 B : gRNA#1 should probably be #3. Please check.
- Maybe detail at least once in the text DCSTAMP, OCSTAMP and DCST.

Response: Thanks for pointing out these shortcomings. We revised our manuscript accordingly to correct these mistakes.

Reviewer #3:

In the manuscript, "Sperm membrane proteins DCST1 and DCST2 are required for the sperm-egg fusion process in mice and fish" by Ikawa et al., the authors described the importance of DCST1 and DCST2 in fusion of sperm with oolemma in vertebrate species. This is a second report in mice and first of its kind in fish. The findings of the manuscript may generate wider interest among the reproductive biologists. The well-designed methodology and clearly drawn interpretation are strength of this paper.

General minor comments:

Comment #18: 1. The consistency in writing sperm/spermatozoa has to be maintained

Response: Thank you for pointing this out. In our original version we had used ‘sperm’ as modifier of a noun (e.g. sperm tail) and spermatozoa as stand-alone word, but we agree with the reviewer that this might be unnecessary complicated. We have therefore revised our manuscript and replaced spermatozoa with sperm throughout.

Comment #19: 2. line 135: DCST2 is indispensable. Agreed. clarity is missing in Line 138, wherein it was mentioned that DCST mutant mice rarely fertilise eggs?. Also, in the subsequent paragraph/results it was reported that sperm from KO (DCST1 and DCST2 genes) males did fuse with eggs, without embryo formation.

Response: We apologize for the insufficient description. Mouse *Dcst1* KO sperm rarely fuse with the oocyte membrane, leading to severe subfertility. Mouse *Dcst2* KO males are completely infertile due to the fusion defect. We corrected the relevant parts as shown below:

Page 3 lines 132-138:

“*Dcst1*^{d1/d1}, *Dcst2*^{d25/d25}, and *Dcst2*^{del/del} male mice successfully mated with female mice.

However, crosses between *Dcst2*^{d25/d25} and *Dcst2*^{del/del} males and wild-type females did not result in any offspring, and crosses with *Dcst1*^{d1/d1} males were only rarely giving rise to pups {pups/plug: 9.01 ± 2.77 [control (Ctrl), 19 plugs], 0.22 ± 0.19 [*Dcst1*^{d1/d1}, 17 plugs], 0 [*Dcst2*^{d25/d25}, 42 plugs], 0 [*Dcst2*^{del/del}, 24 plugs]}, indicating that *Dcst1* mutant males are almost and *Dcst2* males are completely sterile (**Figure 1C**).”

Page 4 lines 156-157: We changed the subheading title “Loss of DCST1/2 causes a sperm-egg fusion defect in mice”.

Page 5 lines 202-205:

“Thus, while acrosome-reacted sperm of *Dcst1* and *Dcst2* KO mice can bind to eggs, they are defective at fusing with the oolemma: KO males of *Dcst1* or *Dcst2* causes a strong impairment or complete loss of sperm-egg fusion, respectively.”

Page 5 line 214: We changed “Sterility” to “Fecundity”.

Comment #20: 3. Materials:

Though the headings are specific, the sub-headings can be clubbed together to provide complete picture. For example, instead of separate heading for antibodies (line 364), this can be clubbed with immunocytochemistry (line 434).

Response: Thank you for your comment. As the same antibodies were used for not only immunocytochemistry but some experiments (e.g., western blotting), we prefer the current form as the heading for antibodies. We are happy to follow the editors’ decision.

Comment #21: In the western and immunocytochemistry loading control and negative control, respectively may be mentioned clearly.

Response: As indicated by the reviewer, we clarified the samples used as the control in the relevant parts in the methods section and figure legends as shown below:

Page 10 lines 424-425: Sperm of *Dcst1*^{d1/wt} and *Dcst2*^{d25/wt} mice were used as the control for sperm motility and IVF.

Page 11 lines 465-466: Sperm of *Dcst1*^{d1/wt} and *Dcst2*^{d25/wt} mice were used as the control.

Page 20 lines 892-894: The band signals of IZUMO1 in TGC and sperm of *Dcst1*^{d1/d1} and *Dcst2*^{d25/d25} male mice were comparable to the control wild-type sperm.

Page 21 lines 923-924: Wild-type sperm were used as the negative control.

Page 21 line 927: The TGC and sperm lysates from Ctrl (*Dcst2*^{wt/wt} and *d25/wt* mice) ...

Comment #22: Line 333: RT-PCR: not clear whether this was realtime PCR or not? Whether MIQE Guidelines were followed while setting up the experiment?

Response: We performed reverse transcription polymerase chain reaction (RT-PCR) to reveal the *Dcst1/2* expression levels in multi-tissues, and quantitative PCR (qPCR) to examine the *Dcst1/2* expression levels in each mutant following MIQE guidelines.

We specified the relevant methods in the material and methods section as shown below (Page 9 lines 391-400 of the revised MS; and Page 2 lines 40-46 of the supplementary information):

“Reverse transcription polymerase chain reaction (RT-PCR) for mouse multi-tissue expression analyses.

Mouse tissues were homogenized in TRIzol reagent (Thermo Fisher Scientific), and total RNA was extracted as described in the instruction manual. Total RNA was reverse-transcribed into cDNA using a SuperScript IV First-Strand Synthesis System (Invitrogen) and Deoxyribonuclease (RT Grade) (Nippon gene). PCR was conducted with primer sets (Table S1) and KOD-Fx neo (TOYOBO). The PCR conditions were initial denaturation at 94°C for 3 minutes, denaturing at 94°C for 30 seconds, annealing at 65°C for 30 seconds, and elongation at 72°C for 30 seconds for 30 or 35 cycles in total, followed by 72°C for 2 minutes.”

“Quantitative PCR (qPCR) to analyze *Dcst1* and *Dcst2* expression levels

The synthesized cDNA (5 ng), primer sets (Table S4), and THUNDERBIRD Next SYBR qPCR Mix (TOYOBO) were used for qPCR. The condition for qPCR was 95°C for 30 seconds, denaturing at 95 °C for 5 seconds, annealing at 65°C for 10 seconds for 40 cycles in total. For melting curve, the samples was treated at 95°C for 15 seconds, followed by increasing the temperature by 0.3 °C from 60°C. *Actb* was used as a reference gene, and the relative difference in the expression level was calculated by the $\Delta\Delta C_t$ method. ”

Other #1: To show the individual data points, we added dots indicating the replicate values in all figures.

Other #2: We added Yonggang Lu as the new author because he performed the revision experiment, and the relevant parts were updated (Page 1 lines 4-24, Pages 15-16 lines 682-721).

REVIEWERS' COMMENTS:

Reviewer #1 (Remarks to the Author):

I am satisfied that the authors have adequately addressed all the comments raised by the reviewers.

Reviewer #2 (Remarks to the Author):

We have revised the paper in its new version, for us it was already acceptable before, it is even more so now.

We have no further comments.

Reviewer #3 (Remarks to the Author):

My comments were satisfactorily addressed. The manuscript is acceptable for publication